


# What could we learn about climate sensitivity from variability in the surface temperature record?

James Douglas Annan[1], Julia Catherine Hargreaves[1], Thorsten Mauritsen[2], and Bjorn Stevens[3]

[1]Blue Skies Research Ltd, The Old Chapel, Albert Hill, Settle, BD24 9HE, UK
[2]Department of Meteorology, Stockholm University, Stockholm, Sweden
[3]Max Planck Institute for Meteorology, Hamburg, Germany

*Correspondence to:* jdannan@blueskiesresearch.org.uk

**Abstract.**

   We examine what can be learnt about climate sensitivity from variability in the surface air temperature record over the instrumental period, from around 1880 to the present. While many previous studies have used the trend in the time series to constrain equilibrium climate sensitivity, it has also been argued that temporal variability may also be a powerful constraint.
We explore this question in the context of a simple widely used energy balance model of the climate system. We consider two recently-proposed summary measures of variability and also show how the full information content can be optimally used in this idealised scenario. We find that the constraint provided by variability is inherently skewed and its power is inversely related to the sensitivity itself, discriminating most strongly between low sensitivity values and weakening substantially for higher values. It is only when the sensitivity is very low that the variability can provide a tight constraint. Our investigations
take the form of "perfect model" experiments, in which we make the optimistic assumption that the model is structurally perfect and all uncertainties (including the true parameter values and nature of internal variability noise) are correctly characterised. Therefore the results might be interpreted as a best case scenario for what we can learn from variability, rather than a realistic estimate of this. In these experiments, we find that for a moderate sensitivity of 2.5°C, a 150 year time series of pure internal variability will typically support an estimate with a 5–95% range of around 5°C (e.g. 1.9–6.8°C). Total variability including
that due to the forced response, as observed in the detrended observational record, can provide a stronger constraint with an equivalent 5–95% posterior range of around 4°C (eg 1.7–5.6°C) even when uncertainty in aerosol forcing is considered. Using a statistical summary of variability based on autocorrelation and the magnitude of residuals after detrending proves somewhat less powerful as a constraint than the full time series in both situations. Our results support the analysis of variability as a potentially useful tool in helping to constrain equilibrium climate sensitivity, but suggest caution in the interpretation of
precise results.

## 1  Introduction

For many years, researchers have analysed the warming of the climate system as observed in the modern instrumental temperature record (spanning the late 19th to early 20th century), in order to understand the response of the climate system to external forcing. For the most part, the focus has been on the long-term warming trend (e.g. Gregory et al., 2002; Otto et al., 2013;





Lewis and Curry, 2015). However, some research has focussed more specifically on the temporal variability exhibited in this temperature record (Schwartz, 2007; Cox et al., 2018a), and this topic is the focus of this paper.

Schwartz (2007) argued on the basis of a simple zero-dimensional energy balance model that an analysis based on the fluctuation-dissipation theorem (Einstein, 1905) could be used to directly diagnose the sensitivity of the Earth's climate sys-
tem $S$ — here conventionally defined as the equilibrium surface air temperature response to a doubling of the atmospheric $CO_2$ concentration — from variability in the observed record of annually and globally averaged surface air temperature observations over the observational record. While we do not wish to repeat the arguments here, we will note that several researchers disputed this analysis, demonstrating *inter alia* that this method did not reliably diagnose the sensitivity of climate models, and also arguing why it could not be expected to do so, given their complexity (Foster et al., 2008; Knutti et al., 2008). Perhaps as
a consequence of these arguments, this line of research was largely ignored for the subsequent decade.

More recently however, Cox et al. (2018a) reopened this question with an analysis based on an emergent constraint approach. That is, rather than following the directly diagnostic approach of Schwartz (2007), they instead observed that a quasi-linear relationship existed across an ensemble of CMIP5 models (Taylor et al., 2012), between the sensitivities of these models, and their interannual temperature variabilities as summarised in a statistic which they denoted $\Psi$. It has been cogently argued that
an emergent constraint should only be taken seriously if supported by some theoretical basis (Caldwell et al., 2014), and Cox et al. (2018a) did indeed present an analysis — again based on simple zero-dimensional energy balance modelling — which qualitatively underpinned this linear relationship. Using the value of $\Psi$ obtained from observations of surface air temperature, together with the empirical relationship between $\Psi$ and $S$ they had derived from the climate models, they produced a best estimate of the equilibrium sensitivity of the climate system of 2.8°C with a likely (66% probability) range of 2.2–3.4°C, a
substantially tighter range than most previous research. However, questions have also been raised about this result (Brown et al., 2018; Rypdal et al., 2018; Po-Chedley et al., 2018)

In this paper, we explore the question of to what extent temporal variability in the globally and annually averaged temperature record can be used to constrain equilibrium climate sensitivity. We consider both the internal variability of the climate system itself, and also the total variability including deviation from a linear trend due to the forced response. Our investigations are
performed in the paradigm of a simple idealised modelling framework, using a two-layer energy balance model which has been widely used to simulate the climate system and which generalises and improves on the performance of the zero-dimensional model. As part of our investigations, we examine the relationship between the $\Psi$ statistic and the equilibrium sensitivity in the model. We also show how the full time series of variability can be used to constrain climate sensitivity, under a variety of idealised scenarios. Our results are based on "perfect model" experiments and therefore may be more readily interpreted as a
best case scenario for what we can learn from variability, rather than a realistic estimate of this.

In the next section, we present the two-layer energy balance model and briefly outline the experimental methods used in this paper. We firstly focus on internal variability, that is to say, the temporal variability arising entirely from to internal dynamics of the climate system in the absence of forcing. We evaluate the power of the $\Psi$ statistic in constraining equilibrium sensitivity, and also consider the more general question of what could in principle be learnt from the full time series. We then consider
variability over the period of the observational record (primarily the 20th century, but with some extension into the 19th and

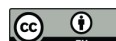



21st centuries). This includes forced variability due to temporal changes in both natural and anthropogenic forcings as well as the internal variability of the climate system. Throughout the paper, the term variability refers simply to all temporal variation in the annually-averaged temperature time series after any linear trend is removed.

## 2 Methods

### 2.1 Model

The basic underpinning of previous work is energy balance modelling of the climate system, from which it is anticipated that interannual variability may be informative regarding the equilibrium sensitivity. While previous research was based on analysis of the simplest possible zero-dimensional single layer planetary energy balance, there is evidence that the behaviour of the climate system over the 20th century is poorly modelled by such a system (e.g. Rypdal and Rypdal, 2014). Therefore, we use here a slightly more complex two-layer model based on Winton et al. (2010). This model has been shown to reasonably replicate the transient behaviour of the CMIP5 ensemble of complex climate models (Geoffroy et al., 2013b, a). The model is defined by the two equations:

$$C_m \frac{dT_m}{dt} = F^t + \lambda T_m - \epsilon \gamma (T_m - T_d) + C_m \delta^t \tag{1}$$

$$C_d \frac{dT_d}{dt} = \gamma (T_m - T_d) \tag{2}$$

This is a two-layer globally-averaged energy balance model which simulates the mixed ($T_m$) and deep ($T_d$) ocean temperature anomalies in the presence of time-varying forcing $F^t$. $\lambda = -F_{2\times}/S$ is the radiative feedback parameter where $S$ is the equilibrium sensitivity and $F_{2\times}$ is the forcing due to a doubling of the atmospheric $CO_2$ concentration. $C_m$ and $C_d$ are the heat capacities of the mixed-layer and deep ocean respectively and $\gamma$ represents the ocean heat transfer parameter. The parameter $\epsilon$ was introduced by Winton et al. (2010) to represent the deep-ocean heat uptake efficacy. In a slight modification to Winton et al. (2010), we add a noise term $\delta^t$ to the first equation to represent the internal variability of the system as was originally introduced in a single layer energy balance climate model by Hasselmann (1976). Here $\delta^t$ is sampled on an annual (ie, time step) basis from a Gaussian $N(0, \sigma)$ where we generally use the value $\sigma = 0.05$ which generates deviations of order 0.05°C on an annual basis. The mixed layer temperature $T_m$ is considered synonymous with the globally averaged surface temperature. The equations are solved via the simple Euler method with a one year time step.

The values of the various adjustable parameters are listed in Table 1. The depths of the mixed and deep ocean layers are used to calculate the heat capacities $C_m$ and $C_d$ respectively based on ocean coverage of 70% of the planetary surface area. The default values for adjustable parameters are given in Table 1 and the values used here lie close to the mean of those obtained by fitting the model to CMIP5 simulations by Geoffroy et al. (2013a). If we set $\gamma = 0$ and ignore the deep ocean then we





| Parameter | Default Value (Prior) | Description |
|:---:|:---:|:---:|
| $S$ | 3.0 (U[0,10]) | Equilibrium climate sensitivity ($^\circ C$) |
| $\lambda$ | 3.7/$S$ | Radiative feedback ($Wm^{-2}\,^\circ C^{-1}$) |
| $\gamma$ | 0.7 (N(0.7,0.2)) | Deep ocean heat uptake parameter ($Wm^{-2}\,^\circ C^{-1}$) |
| $\epsilon$ | 1.3 (N(1.3,0.3)) | Deep ocean heat uptake efficacy |
| $F_{2\times}$ | 3.7 | Forcing of 2×CO$_2$ ($Wm^{-2}$) |
| $D_m$ | 75 | Depth of mixed layer ($m$) |
| $C_m$ | $4.2 \times 10^6 \times 0.7 \times D_m$ | Heat capacity of mixed layer ($Jm^{-2}\,^\circ C^{-1}$) |
| $D_d$ | 1000 | Depth of deep ocean ($m$) |
| $C_d$ | $4.2 \times 10^6 \times 0.7 \times D_d$ | Heat capacity of deep ocean ($Jm^{-2}\,^\circ C^{-1}$) |
| $\sigma$ | 0.05 (N(0.05,0.01)) | Gaussian noise parameter ($^\circ C$) |

**Table 1.** Adjustable parameters and default values

recover the single layer model of Hasselmann (1976) which was used by both Schwartz (2007) and Cox et al. (2018a) in their theoretical analyses.

## 2.2 Bayesian estimation

Our investigations are performed within the paradigm of Bayesian estimation. In general, the Bayesian approach provides us
5   with a way to estimate a set of unknown parameters $\Theta$ from a set of observations $O$ via Bayes' Theorem,

$$P(\Theta|O) = P(O|\Theta)P(\Theta)/P(O). \tag{3}$$

Here $P(\Theta|O)$ is the posterior probability distribution of $\Theta$ conditioned on a set of observations $O$, $P(O|\Theta)$ is the likelihood function that indicates the probability of obtaining observations $O$ for any particular set of parameters $\Theta$, which in this paper will always contain $S$ and may include other parameters. $P(\Theta)$ is a prior distribution for the parameters $\Theta$, and $P(O)$ is
10  the probability of the observations which is required as a normalising constant in the calculation of the posterior probability distribution.

Formally, the value of the observations is fully summarised by the likelihood function $P(O|\Theta)$, but we primarily present our results as posterior pdfs in order to provide an easily interpreted output which can be directly compared to previously published results. We therefore use a uniform prior in $S$ as this is typically the implicit assumption in emergent constraint
15  analyses (Williamson et al., 2019). This choice results in the posterior being visually equivalent to the likelihood even though their interpretation is somewhat different. In some experiments, we will consider that only the sensitivity is unknown, but in others we will consider a wider range of parametric uncertainties. The priors that we use for all uncertain parameters are shown in Table 1.





# 3   Unforced (internal) variability

## 3.1   Using scalar measures of variability to estimate $S$

Schwartz (2007) and Cox et al. (2018a) both summarised the variability in the temperature record with a scalar measure that they argued (based on simple energy balance modelling) should be informative regarding the sensitivity. Schwartz (2007) sum-
marised the variability via the characteristic decorrelation time constant $\tau = \tau(\Delta t) = -\Delta t / \ln(\rho_{\Delta t})$ where $\Delta t$ is an adjustable lag time and $\rho_{\Delta t}$ is the autocorrelation coefficient of the temperature time series at a time lag of $\Delta t$. The method of selecting $\Delta t$ and therefore the estimation of $\tau$ was not presented in an entirely objective algorithmic form, but for the simple one-layer climate model that was considered, the expected value of $\tau$ calculated from a long unforced time series is independent of lag. Cox et al. (2018a) argued that the function $\Psi = \sigma_T / \sqrt{-\ln \rho_1}$ should be linearly related to the equilibrium sensitivity. In
this function, $\rho_1$ is the lag-1 autocorrelation of the time series of annual mean surface temperatures, and $\sigma_T$ is the magnitude of interannual variability of these temperatures. $\Psi$ and $\tau$ are closely related and co-vary very similarly over a wide range of sensitivity when other model parameters are held fixed. Henceforth in this section we focus solely on $\Psi$ as it is more precisely defined and has been recently discussed in some detail (Williamson et al., 2019). However very similar results are also obtained when equivalent experiments are performed using $\tau$.

We now present some investigations into the relationship between $\Psi$ and $S$ in unforced simulations of the two-layer model introduced in Section 2. We perform a multifactorial experiment in which 1000-member ensembles of simulations are integrated for both 150 and 1000 year duration, over a range of $S$ from 0 to 10C, and with $\gamma$ set to either the default value of 0.7 or alternatively set to 0 in which case we recover the single-layer version of the model. All other model parameters are held fixed at standard values in these experiments. Since there is no forced trend in these experiments, we do not include any explicit
detrending step in the analyses.

Figure 1 shows the results obtained when $\Psi$ is calculated from the time series of annual mean surface temperatures produced by these simulations. For 150-year simulations using the single layer model, there is a strong linear relationship between the mean value of $\Psi$ obtained, and the sensitivity of the model, just as Cox et al. (2018a) argued. However, Cox et al. (2018a) did not consider sampling variability, that is to say, the precision with which this expected value of $\Psi$ might be estimated from a
finite time series. As our results show, there is substantial uncertainty in the value of $\Psi$ obtained by individual runs, and there is also strong heteroscedasticity, that is to say, the variance of each ensemble of $\Psi$ values increases markedly with sensitivity. This variation arises from the sequence of noise terms which generate the internal variability in each simulation of the model and is therefore an intrinsic aspect of the theoretical framework relating $\Psi$ to $S$. For these unforced simulations, it seems quite possible for a model with its sensitivity set to a value of 5°C or even higher to generate a time series which has a modest value
for $\Psi$ of say 0.1, even though the expected value of $\Psi$ from such model simulations would be much higher.

When we consider the two-layer model using the standard parameter value of $\gamma = 0.7$ then the situation is a little different. In this case the relationship between sensitivity and $\Psi$ is flatter and more curved, with the expected value of $\Psi$ changing slowly for $S > 4°C$. The underlying explanation for this is quite simple. Any small perturbation to the surface temperature is damped on the annual time scale by a relaxation factor which varies in proportion to $\epsilon\gamma - \lambda$, and $\epsilon\gamma$ is equal to 0.91 for standard





parameter values. Therefore, when $S$ is large, changes in $\lambda = -3.7/S$ have relatively little impact on the total damping and thus both the magnitude and autocorrelation of variability are relatively insensitive to further increases in $S$. Williamson et al. (2019) also presented a theoretical analysis of this two-layer model in which they argued that the response of $\Psi$ was close to linear across the GCM parameter range, and our result confirms this for sensitivity values from around 2 to 4 or even 5°C.

However, the gradual curvature for larger values results in a saturation of the response of $\Psi$ to increases in $S$ and this, together with the increasing sampling uncertainty, has consequences that will be shown in subsequent experiments. Thus our work does not challenge the underlying analysis that they presented, but augments it with additional details.

We now directly consider the question of how useful an observed value $\Psi^o$ can be as a constraint on the equilibrium climate sensitivity through Bayesian estimation. It is not trivial to directly calculate the exact value of the likelihood $P(\Psi^o|\Theta)$ for a

given observed value $\Psi^o$, as $\Psi$ is itself a random variable arising from the stochastic model and thus depends on the sequence of random perturbations that were generated during the numerical integration of the model. Therefore, we use here the technique known as Approximate Bayesian Computation (Diggle and Gratton, 1984; Beaumont et al., 2002). This is a rejection-based sampling technique in which samples are drawn from the prior distribution, used to generate a simulated temperature time series, and rejected if the value of $\Psi$ calculated from this time series does not lie within a small tolerance of the observed value.

The set of accepted samples then approximately samples the desired posterior. We have no observations of long periods of unforced climate variability with the real climate system, so perform a number of synthetic tests in which different hypothetical values for $\Psi^o$ are tested.

Our experiments take the form of a 'perfect model' scenario, where the model is assumed to be a perfect representation of the system under consideration, with no structural imperfections. Our uncertainties here are due solely to unknown parameter

values and internal variability noise. In these experiments, we assume that $\Psi$ for the true system is calculated from a 150 year temperature time series of the unforced system, without any observational error whatsoever. The results of four experiments — using values of $\Psi^o$ which range from 0.05 to 0.2 in regular increments — are shown in Figure 2. There is not necessarily an immediate correspondence between these synthetic values and the observationally-derived value that Cox et al. (2018a) calculated, as we are using unforced model simulations here. Nevertheless, the results are qualitatively interesting. With other

model parameters set to the default values, the four values of $\Psi^o$ used here correspond to the expected value generated by 150 year integrations with sensitivities of approximately 1, 2.5, 5 and 10°C respectively. The figure shows that in this experimental scenario, $\Psi$ can only provide a tight constraint in the first case where the sensitivity is very low. In this case, the 5–95% range of the posterior is an impressively narrow 0.7–1.7°C. For the case $\Psi^o = 0.1$, the equivalent probability interval is 1.8–8.0°C and for higher values of $\Psi$ the posterior is very flat indeed with just the very lowest values of $S$ excluded. Similar results are

obtained when equivalent values of $\tau$ are used as observational constraints.

Strictly, when considering the strength of the constraint, it is the likelihood $P(\Psi|S)$ that we should be considering, rather than the posterior pdf $P(S|\Psi)$, since the latter depends also on the prior. Likelihood values can be directly read off from Figure 2 as the height of the appropriate density curve at specific sensitivities, rather than the integral under it. In the experiment performed with $\Psi^o = 0.1$, the maximum likelihood value is achieved at a value of $S = 2.4$°C, and the likelihood drops by a

factor of 10 at both $S=1.4$°C and $S=7.6$°C. Kass and Raftery (1995) suggest that a likelihood ratio of 10 or more between

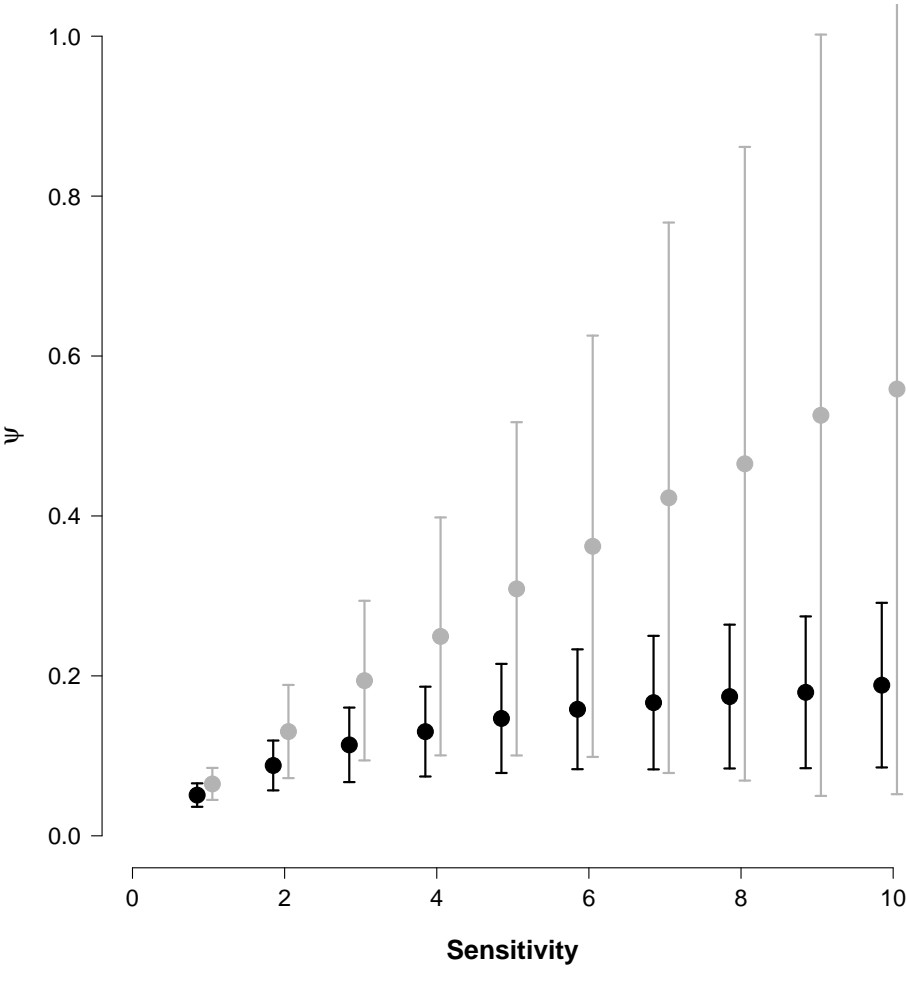

**Figure 1.** Ψ estimated from 150y time series for 1- and 2-layer model. Grey results are for single-layer model and black results for two-layer model. Large dots show means of 1000 simulations, with error bars indicating ±2sd ranges for each ensemble. Results are calculated at each integer value of sensitivity and offset slightly for visibility.

two competing hypotheses could be taken to represent "strong" evidence in favour of one over the other, so if we adopt this linguistic calibration we could say that the observation of $\Psi^o = 0.1$ represents strong evidence in favour of $S = 2.4°C$ versus all values outside of the range 1.4–7.6°C (but conversely, does not represent strong evidence to discriminate between any pair of values inside that range). It is somewhat coincidental that this range seems quite similar to the 5–95% range of the posterior

5  pdf as the philosophical interpretation of the ranges is rather different. There is a strong skew in this range, which extends

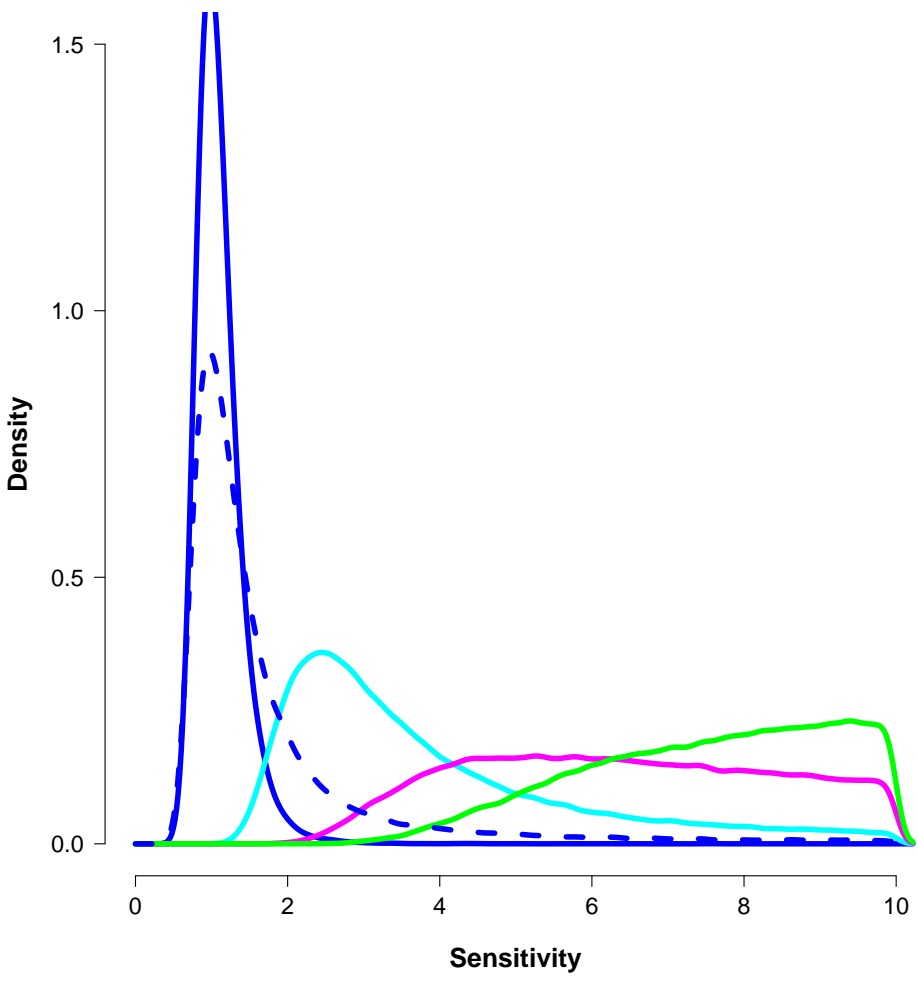

**Figure 2.** Posterior estimates of sensitivity based on $\Psi$ estimated from 150y time series with unforced 2-layer model. Four solid-line pdfs in blue, cyan, magenta and green represent estimates based on $\Psi^o$ = 0.05, 0.1, 0.15, and 0.2 respectively, where only $S$ is uncertain with uniform prior. Dotted blue line represents posterior estimate for $\Psi^o$ = 0.05 with additional modelling uncertainties as described in main text.

much further towards higher values of $S$ than lower ones, compared to the maximum likelihood estimate. We stress that this skew is a fundamental property of the physical model and is not due to the Bayesian analysis paradigm.

## 3.2 Additional uncertainties

The experiments summarised in Figure 2 assume that all model parameters other than $S$ are known with certainty. In reality,
5 we have significant uncertainty as to what values we should assign to several other parameters. We consider just three of these:





the ocean heat uptake parameter $\gamma$, the efficacy or pattern effect parameter $\epsilon$ and the internal noise parameter $\sigma$. Geoffroy et al. (2013a) fitted the two-layer model to various GCM outputs in order to estimate parameter values including $\gamma$ and $\epsilon$ and based on these results we use as priors for these parameters the distributions $N(0.7, 0.2)$ and $N(1.3, 0.3)$ respectively which generously encapsulate their results. Geoffroy et al. (2013a) did not consider internal variability and thus we do not have such

a solid basis for a prior in $\sigma$ and assume a comparable relative uncertainty of 20%, i.e. a prior of $N(0.05, 0.01)$. When we repeat the previous experiments but include these additional parametric uncertainties, then for the experiment where we use $\Psi^o = 0.05$ as a constraint, the posterior for $S$ widens substantially from the previous spread of 0.7–1.7°C, to 0.7–4.8°C as also shown as the dashed blue line in Figure 2. The largest factor generating this substantial increase in uncertainty is due to the uncertainty in $\sigma$. The equivalent posteriors using the larger observational values for $\Psi^o$ also broaden somewhat but this is less

visible in the results as they are of course always constrained by the prior range.

### 3.3    Using the full time series

Although the results in Figure 2 show that an observation of $\Psi$ taken from a short unforced simulation cannot tightly constrain equilibrium sensitivity in this model (except perhaps in the most exceptional of circumstances), it could still be hoped that a more precise constraint could possibly be gleaned by a more advanced analysis that uses some different diagnostic of the time

series. In this section, we show how the total information of the time series can be used. By doing this, we create the most optimistic possible scenario for using internal variability to constrain equilibrium sensitivity of this simple climate model.

     This approach requires us to calculate the likelihood for the full set of observations, $P(O|\Theta)$ where here $O = T_m^i$, $i = 1 \ldots N$ is the full time series of annual surface temperature anomalies. Once the model parameters and forcing are prescribed, the time series of surface temperature anomalies is uniquely determined by the series of random noise perturbations $\delta^i$. Thus, in the

absence of observational error, we can invert this calculation to calculate (up to machine precision) the sequence of annual random noise perturbations $\delta^i$, $i = 1 \ldots N$ that are required in order to replicate any given observed temperature time series. This is why we selected a model time step of one year: with one observation per year, we can precisely invert the model integration to calculate one uncertain noise input per year and thereby reproduce the full model integration to within machine precision. The probability of the model generating the observed sequence is exactly the probability of the required noise

sequence being sampled. This value is readily calculated, since the joint density of these i.i.d. generated $\delta^i$ is simply the product of their individual densities. This unusual approach which we do not believe has been previously implemented in this context is possible here as we are assuming zero observational uncertainty. With this exact likelihood calculation, the Bayesian estimation process is straightforward. In the case where only sensitivity is considered uncertain, it can be performed by direct numerical integration, sampling the sensitivity on a fine regular grid and calculating the likelihood (and therefore posterior)

directly at these values.

     It is worth emphasising that this calculation represents an absolute best case scenario for using the time series of temperature anomalies as a constraint. There can be no diagnostic or statistical summary of the observations that provides more information than the full set of observations themselves contain. Thus, we cannot hope to obtain a better constraint by some alternative analysis of the temperature time series.





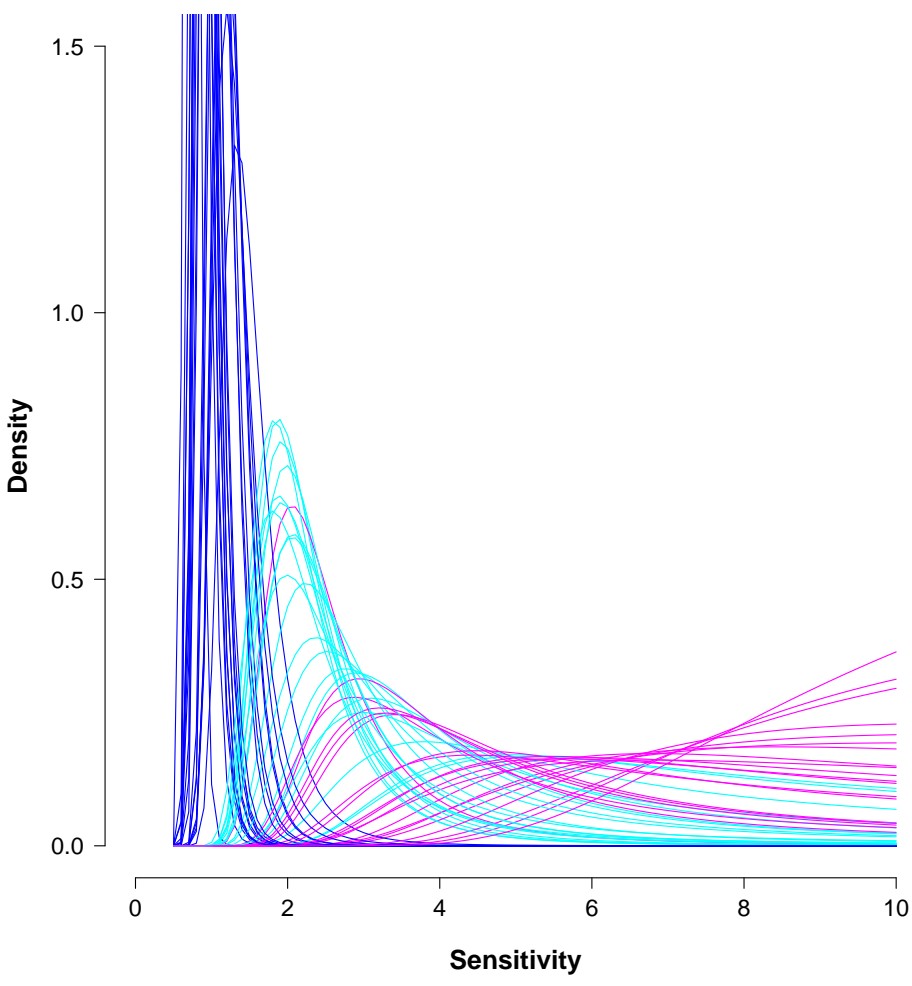

**Figure 3.** Posterior estimates for the climate sensitivity using Bayesian estimation using the full 150 year time series of annual mean surface temperatures. 20 replicates are performed for each true sensitivity of 1, 2.5, 5°C as indicated by the colour blue, cyan and magenta respectively.

Figure 3 shows results obtained from this approach, in the case where only $S$ is considered uncertain. To aid visual comparability with Figure 2, the $y$-axis scale is fixed at the same value despite cutting the peaks of some pdfs. It is not possible to define what a "typical" noise sequence might look like and therefore we plot 20 replications with different randomly generated instances of internal variability for each sensitivity value tested. We can see that the results are generally a little more precise than was obtained using $\Psi$ alone (as shown by the pdfs generally having higher peak densities), though this depends on the specific sample of internal variability that was obtained. It is still only in the case of the lowest sensitivity value of 1°C that





we reliably obtain a tight constraint. With the true sensitivity of 2.5°C, the posterior 5–95% range, averaged over the samples, is 1.9–6.8°C, a little narrower than the 1.8–8°C range obtained when an equivalent $\Psi$ value of 0.1 is used. When additional parametric uncertainties are considered, the constraints again weaken, though not to quite such an extent as in Section 3.2 when only $\Psi$ is used as a constraint. We do not show these results here.

Thus, there appears to be the potential for internal variability, as represented by the full temperature time series, to provide a slightly better constraint than that obtained by a summary statistic alone, but the improvement is marginal and even our optimal calculation which uses the exact likelihood of the full time series cannot accurately diagnose equilibrium sensitivity except when the true value is very low. These results again show a skew similar to that obtained when $\Psi$ was used as the constraint in Sections 3.1 and 3.2. Thus this non-Gaussian likelihood is again an inherent property of the physical model and

not an artefact of the analysis. We mention again that these calculations are made under the three optimistic assumptions that (a) the model is perfect and we have exact knowledge of all other model parameters, (b) we know the forcing to be zero over this time period, and (c) there is no observational error.

## 4    Forced variability

While the theoretical underpinning of Cox et al. (2018a) was originally based on the properties of unforced internal variability,

Cox et al. (2018b) acknowledged that their approach may have benefited from some signature of forced variability entering into their calculations. In order to calculate their $\Psi$ statistic, they applied a windowed detrending method in order to focus on variability of both model simulations and observations of the 20th century. However, the window length of 55 years that they used was justified primarily in empirical terms and cannot remove shorter-term variations in forced response.

In this section, we perform a series of analyses based on 20th century forced simulations, in order to investigate more fully

the potential for such forced effects to improve the constraint. We force the climate model with annual time series for the major forcing factors based on IPCC (Annex II: Climate System Scenario Tables 2013). Our two-layer model with a one-year time step (and Euler method of numerical integration) reacts rather too strongly to short-term spikes in forcing and thus we scale the volcanic forcing to 70% of the nominal value in order to give more realistic simulations. In some of the following experiments, we consider aerosol forcing as a source of uncertainty in addition to that arising from the internal parameters of the model. This

uncertainty is implemented via a scaling factor denoted by $\alpha$ which is uncertain but constant in time, applied to the original aerosol forcing time series. In these cases, our prior distribution for $\alpha$ is $N(1, 0.5)$.

Figure 4 presents 18 simulations from the model, consisting of 5 instances of internal variability for each of three different parameter sets which were chosen to give reasonable agreement with observational data, and additionally one simulation for each of these parameter sets in which internal variability was not included in order to show the pure forced response.

These simulations are shown merely to indicate the typical behaviour of the model under 20th century forcing and are not directly relevant to our analyses. Note that the observations of the real climate system which are also plotted here include observational error (estimated to be roughly $\pm\,0.05$°C at the 1 standard deviation level) whereas the model output is presented as an exact global temperature. Thus it is to be expected that the model results are somewhat smoother and less variable than



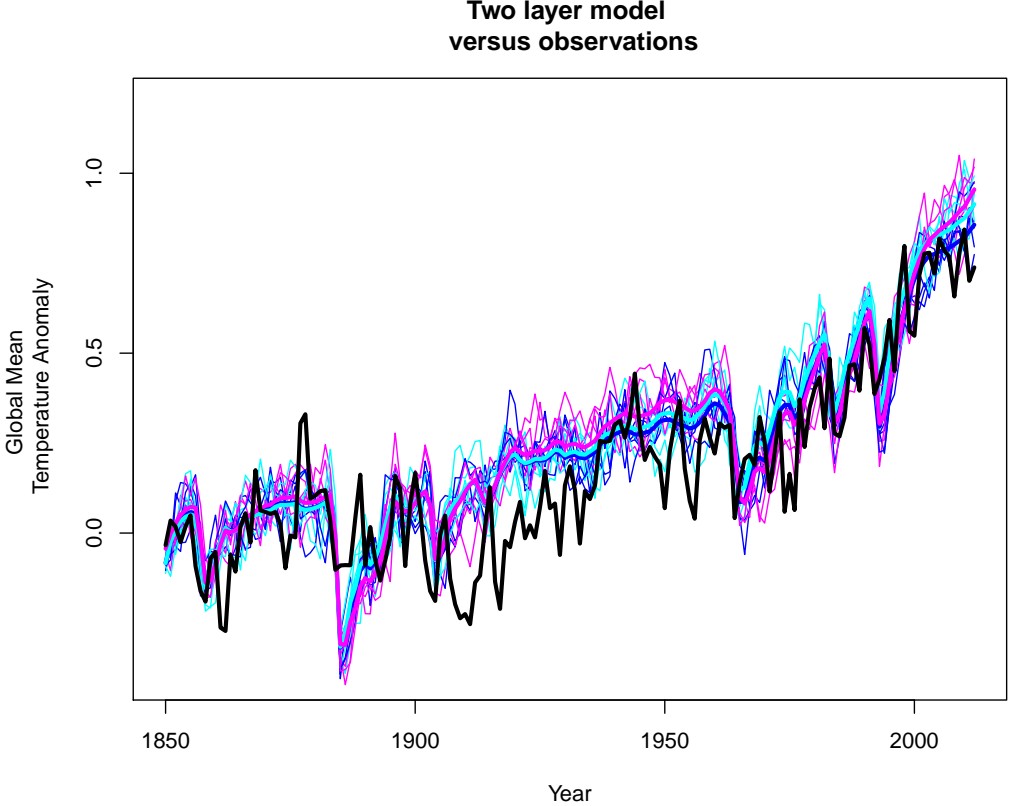

**Figure 4.** Simulations of instrumental period with 2-layer model. Thick lines are forced response excluding internal variability, thin lines are 5 replicates of each parameter set including internal variability. Blue lines: $S = 1.78°C$, $\gamma = 0.7 W m^{-2} {}^\circ C^{-1}$, $\epsilon = 1.3$. Cyan lines: $S = 2.5°C$, $\gamma = 1.0 W m^{-2} {}^\circ C^{-1}$, $\epsilon = 1.7$. Magenta lines: $S = 5°C$, $\gamma = 1.0 W m^{-2} {}^\circ C^{-1}$, $\epsilon = 1.7$, $\alpha = 1.7$. Black line is HadCRUT data.

the observations. The RMS differences between model output and observations is identical to two significant figures for the three simulations without internal variability, at $0.13°C$.

When we hold other parameters at default levels, best agreement between model and data (defined here simply by RMS difference between the two time series) is achieved for a rather low sensitivity of $1.78°C$. If the $\gamma$ and $\epsilon$ parameters are increased slightly above their defaults then we can achieve an equally good simulation (again as measured by RMS difference) with a higher value for sensitivity of $2.5°C$. If, additionally, aerosol forcing is also increased above the default value, then a higher sensitivity still of $5°C$ achieves an equally good match to the observed temperature time series. These simulations help to illustrate why it has been so challenging to effectively constrain equilibrium sensitivity from the long-term observed warming.





## 4.1  Using $\Psi$

In order to assess what we can learn about sensitivity from the variability of 20th century temperature observations, we firstly consider the utility of $\Psi$ calculated from 20th century simulations. Cox et al. (2018a) proposed that the effect of forcing over this interval could be effectively removed by a process of windowed detrending. Figure 5 shows results analogous to those of Figure 1, but using forced simulations of the 20th century rather than unforced control simulations, and with $\Psi$ calculated via the windowed detrending method of Cox et al. (2018a). One very minor discrepancy with the calculations presented in that paper is that our simulations only extend to 2012 (this being the limit of the forcing time series that we are using) and therefore we omit the last 4 years of the time period that they used. This does not significantly affect any of our results.

Grey dots and error bars indicate results obtained when only $S$ is considered uncertain. The values of $\Psi$ obtained for any specific value of $S$ are somewhat larger than those of Figure 1, which supports the claims of Brown et al. (2018) and Po-Chedley et al. (2018) that the windowed detrending has not been wholly effective in eliminating all influence of forcing. The overall nature of the relationship is broadly similar, however. As mentioned previously, results from GCMs indicate significant uncertainty in other model parameters and therefore we have performed additional ensembles of simulations which account for uncertainty both in model parameters and aerosol forcing. These uncertain parameters, and their priors, are the same as in Section 3.2. The combined effect of these uncertainties has little systematic effect on the mean estimate of $\Psi$ but slightly increases the ensemble spread. Interestingly, in contrast to our earlier experiments, no single factor appears to have a dominant effect here.

Black crosses also shown on this figure show results obtained from the CMIP5 ensemble. We obtained historical simulations performed by 23 CMIP5 models from the Climate Explorer website (https://climexp.knmi.nl/). Where multiple simulations were performed with a single model, we show all results (amounting to 89 model runs in total) and these vary substantially due solely to the sample of internal variability in each simulation. The CMIP5 models appear to generate slightly lower values of $\Psi$ than the two-layer model does with the same sensitivity, although the two sets of results seem broadly compatible. Reasons for the difference may include biases in parameters of the two-layer model, structural limitations, or differences in the forcings used. Although we are not replicating the emergent constraint approach here, we do note that there is a significant correlation in the CMIP5 ensemble results shown here between their sensitivities and $\Psi$ values. However, the relationship seen here is is weaker than that obtained by Cox et al. (2018a) for a different (though overlapping) set of models, and explains a lower proportion of the variance. This remains true whether we perform the regression with $S$ as predictor, as suggested by our Figure, or when using $\Psi$ as predictor as in Cox et al. (2018a).

Figure 6 shows results generated when various hypothetical values for $\Psi^o$ are used to constrain model parameters for 20th century simulations. As before, we test sensitivity values of 1, 2.5 and 5°C which here correspond to values for $\Psi^o$ of 0.1, 0.18, and 0.25. As in the previous experiments, the smallest value of $\Psi^o$ generates an impressively tight constraint with a 5–95% range of 0.8–1.7°C when only $S$ is considered uncertain. This grows to 1.8–7.5°C for $\Psi^o = 0.18$ and the larger value of $\Psi^o = 0.25$ provide very little constraint. When the additional parametric and forcing uncertainties are considered, the tightest range corresponding to $\Psi^o = 0.1$ grows a little to 0.8–1.9°C and the $\Psi^o = 0.18$ case spreads to 1.8–8.8°C.



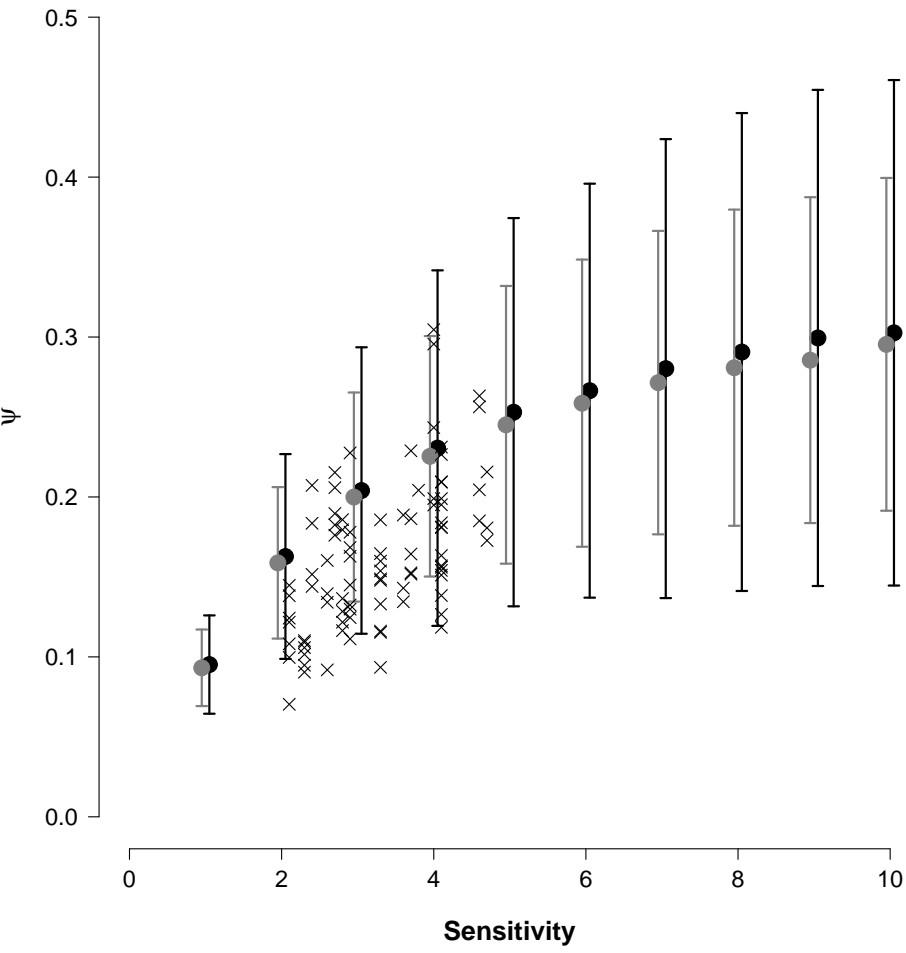

**Figure 5.** $\Psi$ estimated from 20th century simulations from 2-layer model. Grey results are based on simulations where only $S$ is considered uncertain. Black results additionally account for uncertainty in $\gamma$, $\epsilon$, $\sigma$ and $\alpha$. Large dots show means of 1000 simulations, with error bars indicating $\pm 2$sd ranges for each ensemble. Crosses indicate results generated by CMIP5 models.

When we use the observational value of $\Psi^o = 0.13$ (calculated from HadCRU data) and include multiple parametric uncertainties, the 17–83% posterior range is 1.3–2.6°C and the 2.5–97.5% range is 1.0–5.5°C. These ranges are somewhat larger than the equivalent results presented by Cox et al. (2018a) which were 2.2–3.4°C and 1.6–4.0°C respectively, despite the highly optimistic perfect model scenario considered here.

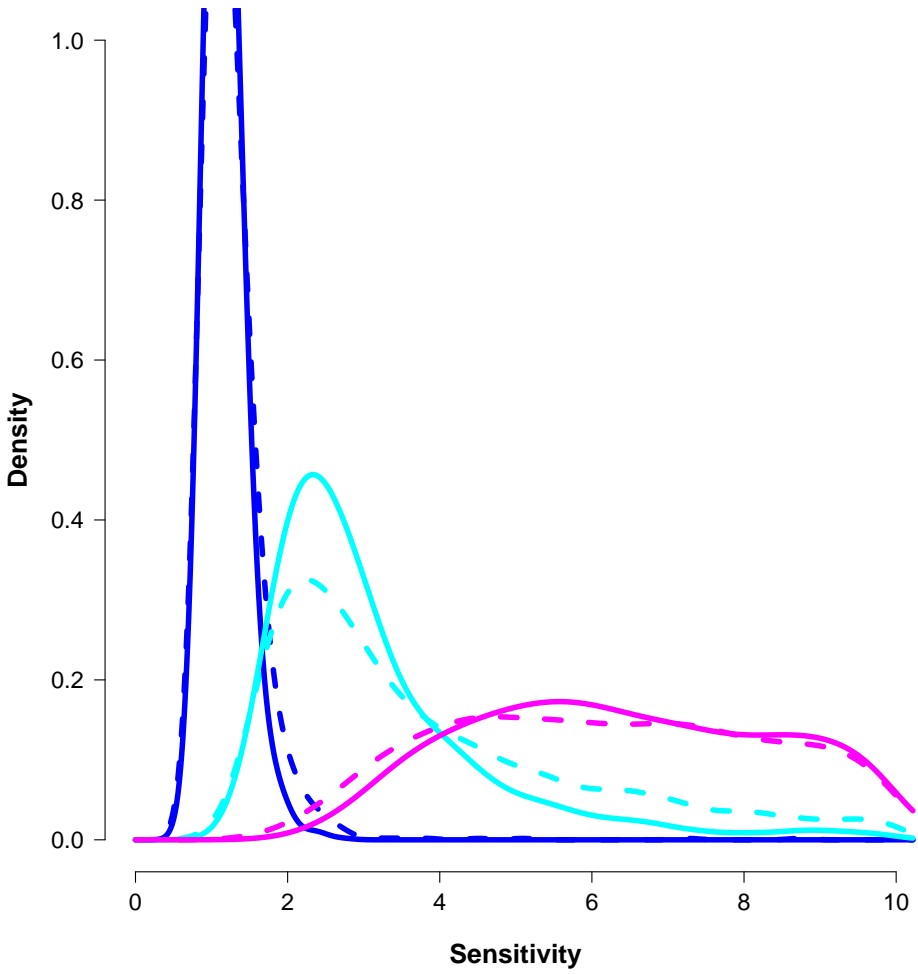

**Figure 6.** Posterior estimates of sensitivity based on $\Psi$ estimated from 20th century simulations with 2-layer model. Three solid-line pdfs in blue, cyan and magenta represent estimates based on $\Psi^o = 0.1, 0.18$, and $0.25$ respectively, where only $S$ is uncertain with uniform prior. Dashed lines represent equivalent posteriors with additional modelling uncertainties as described in main text.

## 4.2 Using the full time series

Finally, we repeat the approach of Section 3.3, and use the full information of the time series, by the same method of inverting the model to diagnose the internal variability noise that is required to generate the observed temperature time series. Since we are interested solely in variability, we only consider the temperature residuals after detrending. We use a simple linear de-
5   trending over the period 1880-2012, which will leave a signature primarily due both to volcanic events and also the contrasting





temporal evolution of negative aerosol forcing and positive greenhouse gas forcing, which both generally increase throughout this period but which exhibit different multidecadal patterns.

The calculation is similar to that of Section 3.3, but there are some minor details which are worth mentioning. Although the detrending is performed over the interval 1880-2012, we initialise the simulations in 1850 to allow for a spin-up. In contrast to
Section 3.3 where detrending was not performed, knowledge of the residuals after detrending does not actually enable an exact reconstruction of the internal variability of the model simulation, as any random trend in this internal variability will have been removed by detrending. In fact a whole family of different model simulations will be aliased onto the same residuals. Therefore our inversion only truly calculates the noise perturbations after the removal of the component that generates any linear trend. This is not a significant problem for the likelihood calculation as the effect of this aliasing is minor and its dependence on
model parameters is negligible.

The results of multiple replicates are shown in Figures 7 and 8, in which we consider firstly the case where only $S$ is uncertain, and then our larger set of parametric uncertainties. In the case where only $S$ is unknown, the full time series of detrended residuals provides strong evidence on $S$ which can as a result be tightly constrained except perhaps when it takes a high value such as 5°C. For $S = 2.5$°C, the posterior 5–95% range is typically under 2C in width, with the average of our
samples being 1.9–3.5°C. When multiple uncertainties are considered, the constraint is however markedly weakened. For a true sensitivity of $S = 2.5$°C, the posterior $5 - 95\%$ range is typically around 4 degrees, at $1.7 - 5.6$°C. As in the unforced experiments, these optimal constraints are clearly narrower than can be obtained by using the $\Psi$ statistic, but are not necessarily tight enough to be compelling in themselves.

## 5   Conclusions

We have explored the potential for using interannual temperature variability in estimating equilibrium sensitivity. While — as Williamson et al. (2019) argued — there is generally a quasi-linear relationship between $S$ and the expected value of $\Psi = \sigma_T / \sqrt{-\ln \rho_1}$ over a reasonable range of $S$ in the simple energy balance model, this relationship saturates for higher $S$ and furthermore, sampling variability is significant and highly heteroscedastic. These properties undermine the theoretical basis for the linear regression emergent constraint approach which was presented by Cox et al. (2018a), as the ordinary least squares
regression method relies on a linear relationship with homoscedastic errors. The behaviour of the model instead results in an inherently skewed likelihood $P(\Psi|S)$ with a long tail to high values for $S$. Thus, while $\Psi$ statistic can indeed be informative on $S$, the constraint it provides based on internal variability in the case of unforced simulations is rather limited. We have shown how it is possible in principle to extract the full information from time series of annual temperatures, by calculating the exact likelihood for the complete set of these observations. However, even in this scenario of a perfect model with a few
well-characterised parametric uncertainties and no observational uncertainty on the temperature time series, the constraint on sensitivity is seriously limited by the variability inherent to the model. It is only in the case where the true value of the sensitivity is very low that such an approach can generate a tight constraint. For example, if the true sensitivity takes a moderate value of 2.5°C, then we could only expect to generate a constraint with a typical 5–95% range of around 1.9–6.8°C. As was the case





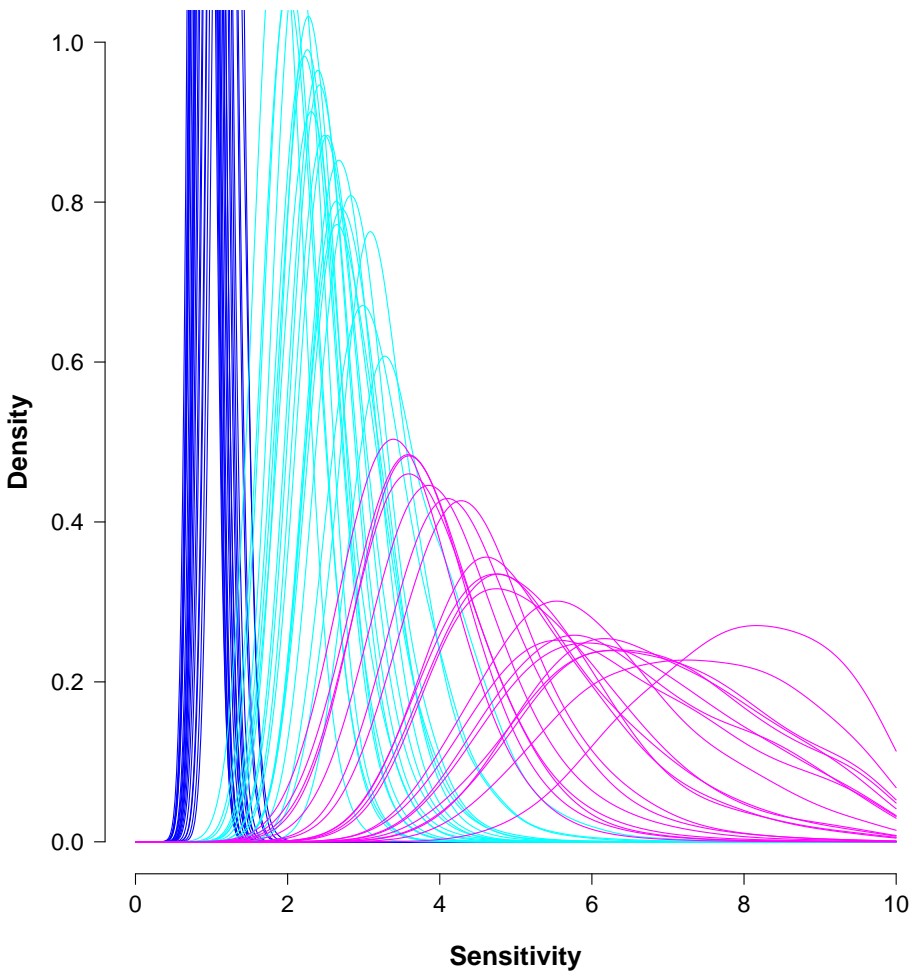

**Figure 7.** Posterior estimates for the climate sensitivity using Bayesian estimation using the full 1880–2012 observational time series of annual mean surface temperature anomalies after detrending. 20 replicates are performed for each true sensitivity of 1, 2.5, 5°C as indicated by colours blue, cyan and magenta respectively. Only $S$ is considered uncertain.

when using $\Psi$, estimates generated from the full time series are rarely close to symmetric and instead are typically skewed with a long tail to high values. This skew is an inherent property of the physical model that defines the likelihood and not an artefact of our analysis methods.

Forced variability, such as that occurring during the instrumental period, does provide additional information in our experi-
5 ments, and therefore we could in theory hope to calculate a narrower posterior range, with a typical width of around 4°C (e.g. 1.7–5.6°C) when the true sensitivity is 2.5°C. It must however be emphasised that these calculations rely on very optimistic

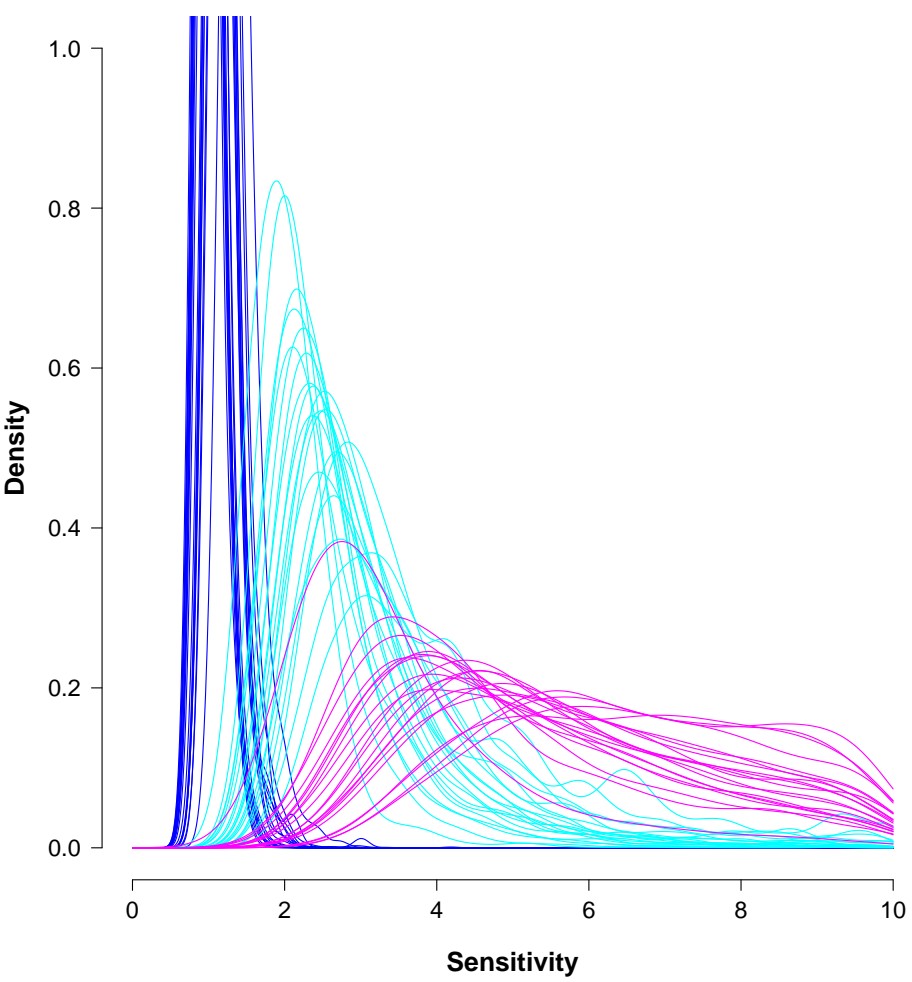

**Figure 8.** As Figure 7 but for multiple parametric uncertainties.

assumptions and therefore represent a best case that is unlikely to be realised in reality. Nevertheless, our results do suggest that variability can inform on the sensitivity and may generate a useful constraint in addition to that arising from the longer-term observed trend.

# 6 Acknowledgements

5  TM acknowledges funding from the European Research Council (ERC) (Grant agreement No.770765) under the European Union's Horizon 2020 research and innovation program (Grant agreement No.820829).



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
