# Peer review of "What could we learn about climate sensitivity from variability in the surface temperature record?"

_Earth System Dynamics, 2019_

## Short Comment (SC1) · 17 Jan 2020

According to the Maximum Entropy Principle, the variance goes as the square of the mean if all one has is knowledge of the mean. So as any estimate of the climate sensitivity increases, the variance will increase as well. Moreover any uncertainty one has in climate sensitivity is reflected by this variance. That's what I am getting out of the paper considering no mention is made of the major source of natural variability in the surface temperature record – ENSO.

---

## Short Comment (SC2) · 4 Feb 2020

1. Page 5 line 9: the Cox, Huntingford and Williamson (2018) assertion that $\Psi = \sigma T /\sqrt{-\ln \alpha\_1}$ (here using rho_1 rather than the original $\alpha\_1$ to denote the year-one-lag autocorrelation) is linearly related to the equilibrium climate sensitivity S is adopted. Cox et al. derived that relationship using an one-box model with no representation of the deep ocean, which is generally viewed as unrealistically simple.

Williamson, Cox and Nijsse (2019) subsequently extended the Cox, Huntingford and Williamson analysis to a more realistic 2-box model, as used in this paper except in their case without the deep ocean heat uptake efficacy parameter ($\varepsilon$). However, the

original Cox, Huntingford and Williamson assertion about $\Psi$ is fairly obviously incorrect for a 2-box model. The Williamson, Cox and Nijsse analysis is made on the valid basis of the slow time constant of the two box model being much longer than the fast time constant, so that the fast mode dominates the variance and autocorrelation. Combining their equation (22) with the definition of $\Psi$, gives $\Psi \approx \sigma\_T \sqrt{\tau\_f}$. Using this relation to substitute for $\tau\_f$ in their equation (21) then yields $\Psi \approx (\sigma\_Q/\sqrt{2})$ a_f/$\lambda$. They then use the relation ECS = Q_2xCO2 / $\lambda$ to obtain from this ECS $\approx (\sqrt{2}$Q_2xCO2/$\sigma\_Q$ a_f) $\Psi$, on the basis that a_f is unrelated to $\lambda$.

However, as Williamson, Cox and Nijsse correctly state, over the CMIP5 range of [fitted 2 box model] parameters a_f $\approx \lambda/(\lambda + \gamma)$. Regression analysis across 34 CMIP5 models gives an R-squared of 0.996 for this approximation to a_f, with an intercept of $-0.03$ and a slope coefficient of 1.03. In the context of a 2-box-$\varepsilon$ model with an added deep ocean heat uptake efficacy parameter $\varepsilon$, this approximation becomes af $\approx \lambda/(\lambda + \varepsilon\gamma)$. Regression analysis across 34 CMIP5 models gives an R-squared of 0.998 for this approximation to a_f, with an intercept of $-0.02$ and a slope coefficient of 1.015.

Accordingly, for the 2-box-$\varepsilon$ model used, substituting af $\approx \lambda/(\lambda + \varepsilon\gamma)$ in a_f/$\lambda \approx \sqrt{2}$ $\Psi/\sigma\_Q$ one gets $1/(\lambda + \varepsilon\gamma) \approx \sqrt{2}$ $\Psi/\sigma\_Q$. Multiplying by Q_2xCO2 implies that $\sqrt{2}$ Q_2xCO2 $\Psi/\sigma\_Q \approx$ Q_2xCO2/$(\lambda + \varepsilon\gamma)$.

However, Q_2xCO2/$(\lambda + \varepsilon\gamma)$ represents a measure of short term transient climate response, not of equilibrium response.

When regressing Q_2xCO2/$(\lambda + \varepsilon\gamma)$ on ECS and TCR, both as estimated from the fitted 2-box-$\varepsilon$ model, across 34 CMIP5 models, the R-squared is 0.983 but the coefficient on ECS is very small $(-0.04)$ and significant only at the 3% level. When regressing without an intercept term (which appears more appropriate) the R-squared is 0.999, the coefficient on TCR is $1.04 \pm 0.03$ but the coefficient on ECS is $-0.03$ and insignificant even at the 10% level.

This analysis implies that in a 2-box-$\varepsilon$ model, and in AOGCMs insofar as their behaviour

can be fit by that model, $\Psi$ is mathematically (and indeed physically) linearly related to TCR, and only indirectly related to ECS. It is accordingly unsound to regard $\Psi$ as an emergent constraint on ECS save through the relationship between ECS and TCR, irrespective of the strength of the correlation between $\Psi$ and ECS in a particular model ensemble, at least insofar as it is valid to use a 2-box model.

In CMIP5 AOGCMs $\Psi$, as derived from locally detrended piControl simulation data, does nevertheless appear to be more strongly correlated with ECS than with $1/(\lambda + \varepsilon\gamma)$, all as estimated from a fitted 2-box-$\varepsilon$ model. That suggests that a 2-box-$\varepsilon$ model is poor at representing AOGCM behaviour on the inter-year timescales relevant to $\sigma\_T$ and rho_1, possibly because near surface temperature over land, sea ice and shallow water responds much more quickly to forcing than does temperature over the open ocean.

This analysis throws doubt on the validity of assessing what can be learnt about equilibrium, as opposed to transient, climate sensitivity from variability in the surface temperature record in a 2-box-$\varepsilon$ model, and potentially also on the physical basis for the related emergent constraint on ECS.

2. Page 6 lines 31-35 & page 7 lines 1-5 : the authors quite rightly use the likelihood ratio rather than the Bayesian posterior density to infer the strength of the constraint. However, the use of a likelihood ratio (Bayes Factor) of 10 or more as suggested by Kass and Raftery (1995) for comparing two competing point hypotheses seems unduly demanding here, where the aim is instead to derive an uncertainty range. A 5-95% uncertainty range can be derived directly from the likelihood ratio, on the assumption that - as is commonly the case - the likelihood function approximately follows a normal distribution or a one-to-one transformation of the parameter exists under which it would do so (e.g., Pawitan et al 2001). The simplest way of doing so is to use the signed root log-likelihood ratio (SRLR), as was done in Allen et al (2009). The SRLR method implies that a 5-95% range spans likelihoods that exceed 0.258 times the maximum likelihood, a likelihood ratio of 3.87, much lower than 10. Uncertainty in additional

parameters can be allowed for by considering all parameters jointly and applying the SRLR method to the profile likelihood.

Also three small points:

3. Page 3 line 20: the model in equations (1) and (2) is not the Winton et al (2010) model: it is the Held et al (2010) model. The Winton et al model, although similar to the Held et al. model, has a basic difference in that its efficacy parameter $\varepsilon$ applies to total ocean heat uptake, and is found to vary significantly over time in AOGCMs, whereas in the Held et al model $\varepsilon$ only applies to deep ocean heat uptake, as in equation (1), and is found to fit AOGCM behaviour with (as here) a constant $\varepsilon$ value. The cited Geoffroy et al (2013a) paper uses the Held model, not the Winton model.

4. Page 4 Table 1: there is a sign error in the default value for the radiative feedback parameter, $\lambda$. The way this parameter is used in equation (1) implies it is negative, but Table 1 defines it as 3.7/S not $-3.7$/S.

5. Page 13, line 30: the title of Kass and Raftery's 1995 paper is just "Bayes Factors".

Nicholas Lewis

References

Allen, M. R., D. J. Frame, C. Huntingford, C. D. Jones, J. A. Lowe, M. Meinshausen, and N. Meinshausen, 2009. Warming caused by cumulative carbon emissions towards the trillionth tonne. Nature, 458, 1163–1166, doi:10.1038/nature08019.

Held, I.M., Winton, M., Takahashi, K., Delworth, T., Zeng, F. and Vallis, G.K., 2010. Probing the fast and slow components of global warming by returning abruptly to preindustrial forcing. Journal of Climate, 23(9), pp.2418-2427.

Pawitan, Y., 2001: In All Likelihood: Statistical Modeling and Inference Using Likelihood. Oxford University Press, 514 pp.

---

## Short Comment (SC3) · 4 Feb 2020

The Nicholas Lewis comment is nonsense. Thermal uptake by the ocean is a diffusional behavior and diffusion is only approximated by an infinite-box model corresponding to the diffusion equation. I would ignore his suggestions apart from the typos he found.

---

## Editor Comment (EC1) · Steven Smith (Editor) · 5 Feb 2020

I would like to remind all commenters to keep their communications professional and to stick to technical and scientific aspects of the paper and any posted comments. If you disagree with something, please relate the technical argument you might wish to make. (S Smith, handling editor.)

---

## Short Comment (SC4) · 7 Feb 2020

I am gratified to see a resurgence of interest in the approach of inferring the sensitivity of global mean surface temperature anomaly T to sustained forcing from fluctuations in time series of T that is reflected in this manuscript and also in Cox et al. (2018) and Williamson et al. (2018). However I note some concerns with this manuscript as well as the two earlier papers. Unfortunately there is some sloppiness in definitions in the earlier papers that carries over to the present paper that makes it tough to evaluate the quantity $\Psi$ and interpret the inferences drawn from that quantity. I also have some concerns over the treatment of the two compartment system that the authors may wish

to take into consideration. At the end of the day, however, as the finding of the present manuscript is that the determination of the climate sensitivity by the approach of Cox et al. (2018) fails, perhaps some of these niceties are of secondary importance. Still, one might hope for more attention to detail.

In conclusion, despite the concerns raised here, the authors are to be applauded for revisiting the question of whether fluctuations in the observed record of global mean surface temperature can usefully constrain climate sensitivity. It seems as if the answer is no, at least so far, but I am not sure that the last word has been written.

Please also note the supplement to this comment:
https://www.earth-syst-dynam-discuss.net/esd-2019-90/esd-2019-90-SC4-supplement.pdf

**Supplement:**

Stephen E. Schwartz

ses@bnl.gov

Received and published XX February, 2020

I am gratified to see a resurgence of interest in the approach of inferring the sensitivity of global mean surface temperature anomaly $T$ to sustained forcing from fluctuations in time series of $T$ that is reflected in this manuscript and also in Cox *et al*. (2018) and Williamson *et al*. (2018). However I note some concerns with this manuscript as well as the two earlier papers. Unfortunately there is some sloppiness in definitions in the earlier papers that carries over to the present paper that makes it tough to evaluate the quantity $\Psi$ and interpret the inferences drawn from that quantity. I also have some concerns over the treatment of the two compartment system that the authors may wish to take into consideration. At the end of the day, however, as the finding of the present manuscript is that the determination of the climate sensitivity by the approach of Cox *et al*. (2018) fails, perhaps some of these niceties are of secondary importance. Still, one might hope for more attention to detail.

First, to the definition of $\Psi$ (page 5, line 9)

$$\Psi = \frac{\sigma_T}{(\ln \rho_1)^{1/2}} .\qquad(1)$$

$\Psi$ is important as, at least as argued by Cox *et al*. (2018) and as examined in the present manuscript' in that it forms the basis for determination of Earth's equilibrium climate sensitivity without requirement of knowledge of forcing as

$$\lambda = \frac{\sigma_T}{\sigma_Q}\left(\frac{\ln \rho_1}{2}\right)^{1/2}\qquad(2)$$

or, in terms of $\Psi$,

$$\lambda^{-1} = \frac{2^{1/2}}{\sigma_Q} \Psi \tag{3}$$

or in terms of equilibrium sensitivity $S_{eq}$,

$$S_{eq} \equiv F_{2\times CO2}\lambda^{-1} = 2^{1/2}\frac{F_{2\times CO2}}{\sigma_Q}\Psi \ . \tag{4}$$

I am not sure I see $\sigma_T$ and $\sigma_Q$ explicitly defined in the present manuscript; presumably they are the standard deviations of the respective quantities (square root of variance) over a time window, and presumably after linear detrending. It is explicitly stated by Cox $et\ al.$ (2018) that $\sigma_T^2$ and $\sigma_Q^2$ are the variances of the time series of $T$ and forcing, calculated for windows of width 55 yr, after linear detrending in a given time window. For a stationary quantity, $\sigma_T$ would be a constant over the time series, with only the uncertainty in $\sigma_T$ being a function of the time window chosen to evaluate the quantity, decreasing with increasing time window (as the square root). However for a nonstationary time series whose mean value over a time window changes with time, $\sigma_T$ calculated relative to the mean of the time series over that window depends on the location and width of the time window, generally increasing with increasing width. Further, it would seem to be essential to specify whether $\sigma_T$ is calculated relative to the mean of the time series in the window or relative to the detrending function (of which a linear function is only one of a myriad of possible detrending functions). The above considerations apply also to $\sigma_F$. Similar considerations attach also to definition of the lag-1-year autocorrelation coefficient $\rho_1$. It thus seems essential that the authors explicitly state how these quantities are determined. Finally, it would seem that a proponent of this method would need to establish that $\Psi$ approaches some sort of limiting value as a function of increasing (or decreasing) window width; if that is not the case then the approach must be considered suspect for that reason alone as well as for other reasons adduced in the present manuscript.

That said, there remains question about Eq (2) itself on which the present manuscript relies. Clearly the derivation given by Cox $et\ al.$ (2018) must be viewed as flawed, the quantities on the left hand sides of each of the equations (Eqs 4 and 5 of that paper) leading to Eq (1) above having different dimension from the respective right hand sides of those equations. So there is a need for a convincing derivation of Eq (1) if it is to be the cornerstone of analyses such as this, generally, and, in particular, of the present manuscript.

On the subject of dimension of quantities, it is clear from Eq (1) above that the dimension of $\Psi$ is temperature (unit K or °C). In the present manuscript (e.g., p. 7, line 2; Figure 1) it is presented as a dimensionless quantity without unit, although I note the unfortunate absence of units in axis labels of many of the figures.

Staying with the relation of $\Psi$ to climate sensitivity, as the fluctuations that are used to determine $\Psi$ are characteristic of the upper ocean, it would seem that for the two compartment model used by Annan *et al.*, the pertinent climate sensitivity quantity would be the transient sensitivity, not the equilibrium sensitivity.

I turn now to Equations 1 and 2 of Annan *et al.* used in their model calculations. First I would question why the efficacy term is introduced (other than that it was introduced by Winton *et al.* (2010) and the present authors have uncritically accepted it). If one defines $\gamma' \equiv \varepsilon\gamma$ , $z'_d \equiv \varepsilon z_d$ , and $C'_d \equiv \varepsilon C_d = \varepsilon D_d c_{vsw} = D'_d c_{vsw}$ , where $D'_d \equiv \varepsilon z_d$ , then Equations 1 and 2 describing the change in heat content in the two compartments become

$$
\begin{aligned}
C_m \frac{dT_m}{dt} &= F(t) - \lambda T_m - \gamma'(T_m - T_d) + C_m \delta(t) \\
C'_d \frac{dT_d}{dt} &= \gamma'(T_m - T_d)
\end{aligned}
\tag{5}
$$

identical to the equations in Schneider and Thompson (1981), Gregory (2000), Held *et al.* (2010), the only difference being the magnitudes of the transfer coefficient and the depth of the deep ocean compartment. Importantly, however, as written in this way the equations exhibit equal piston velocity in both directions and conserve energy between the two compartments, as they must. Also there are only two adjustable parameters, $\gamma'$ and $D'_d$ instead of the original three. The original and revised parameters corresponding to the several traces in Figure 4 of Annan *et al.* are listed in Table 1. The revised parameters clearly display the changes in the several parameters in the several traces. The difference between Blue and Cyan is increase in equilibrium sensitivity being compensated by increase in transfer coefficient $\gamma'$ from the mixed layer ocean to the deep ocean; the change in the depth of the deep ocean $D'_d$ confounds the comparison but it is of very minor importance because of the low rate of return of heat from the deep ocean to the mixed layer. The difference between Cyan and Magenta shows, for fixed transfer coefficient, the compensation of the increase in $\alpha$, the multiplier on aerosol forcing by

increase in equilibrium sensitivity $S_{eq}$, where $S_{eq} \equiv F_{2\times CO2}\lambda^{-1}$ in the governing equations, similar to that shown for example by Tanaka and Raddatz (2011) and Schwartz (2018). It is, of course, this compensation, together with the large uncertainty in aerosol forcing, that motivates the search for an alternative approach to determination of sensitivity as undertaken by Schwartz (2007), Cox *et al.* (2018), and the present manuscript.

**Table 1**. Parameters for Figure 4 of Annan *et al.* as originally presented (without ') and as re-expressed here (with ').

| Trace color | $S_{eq}$ K/2× | $\alpha$ | $\gamma$ W m$^{-2}$ K$^{-1}$ | $\varepsilon$ | $D_d$ m | $C_d$ J m$^{-2}$ K$^{-1}$ | $\gamma'$ W m$^{-2}$ K$^{-1}$ | $C'_d$ J m$^{-2}$ K$^{-1}$ | $D'_d$ m |
|---|---|---|---|---|---|---|---|---|---|
| Blue | 1.78 | 1.0 | 0.7 | 1.3 | 1000 | 2.94E+09 | 0.91 | 3.82E+09 | 1300 |
| Cyan | 2.5 | 1.0 | 1.0 | 1.7 | 1000 | 2.94E+09 | 1.7 | 5.00E+09 | 1700 |
| Magenta | 5 | 1.7 | 1.0 | 1.7 | 1000 | 2.94E+09 | 1.7 | 5.00E+09 | 1700 |

The transfer coefficient $\gamma'$, while rather uncertain, is constrained considerably by observations of the increase in ocean heat content over the past half century. Based on measurements of increase of global temperature and ocean heat content as assessed in AR5 (Hartmann *et al.*, 2013; Rhein *et al.*, 2013), the heat transfer coefficient from a mixed layer of depth 75 m to the deep ocean $\gamma$ is constrained to $0.73 \pm 0.20$ K W m$^{-2}$ (one sigma). The values of $\gamma$ in the examples given by Annan *et al.* thus range from the high end to well beyond the high end of this observationally constrained range. The authors might wish to consider this constraint in the examples they present.

In conclusion, despite the concerns raised here, the authors are to be applauded for revisiting the question of whether fluctuations in the observed record of global mean surface temperature can usefully constrain climate sensitivity. It seems as if the answer is no, at least so far, but I am not sure that the last word has been written.

**References**

Cox, P.M., Huntingford, C. and Williamson, M.S., 2018. Emergent constraint on equilibrium climate sensitivity from global temperature variability. *Nature*, 553, 319-322. https://www.nature.com/articles/nature25450.

Gregory, J. M. (2000), Vertical heat transports in the ocean and their effect on time-dependent climate change, *Climate Dynamics*, *16*(7), 501-515, doi:10.1007/s003820000059.

Hartmann, D. L., *et al.* (2014), Observations: Atmosphere and Surface, in *Climate Change 2013 – The Physical Science Basis: Working Group I Contribution to the Fifth Assessment Report of the Intergovernmental Panel on Climate Change*, edited by T. F. Stocker *et al.*, Cambridge University Press, Cambridge.

Held, I. M., M. Winton, K. Takahashi, T. Delworth, F. Zeng, and G. K. Vallis (2010), Probing the Fast and Slow Components of Global Warming by Returning Abruptly to Preindustrial Forcing, *J. Climate*, *23*, 2418-2427, doi:10.1175/2009JCLI3466.1.

Rhein, M., *et al.* (2013), Observations: Ocean, in *Climate Change 2013 – The Physical Science Basis: Working Group I Contribution to the Fifth Assessment Report of the Intergovernmental Panel on Climate Change*, edited by T. F. Stocker *et al.*, Cambridge University Press, Cambridge.

Schneider, S. H., and S. L. Thompson (1981), Atmospheric $CO_2$ and climate: Importance of the transient response, *J. Geophys. Res. Oceans*, *86*, 3135-3147, doi:10.1029/JC086iC04p03135.

Schwartz S. E. (2007) Heat capacity, time constant, and sensitivity of Earth's climate system. J Geophys Res 112(D24):D24S05. doi:10.1029/2007JD008746

Schwartz, S. E. (2018), Unrealized Global Temperature Increase: Implications of Current Uncertainties, *J. Geophys. Res. Atmos.*, *123*, 3462-3482, doi:10.1002/2017JD028121.

Winton, M., Takahashi, K. and Held, I. M., 2010. Importance of ocean heat uptake efficacy to transient climate change. *J. Climate*, **23**, 2333-2344. https://journals.ametsoc.org/doi/abs/10.1175/2009JCLI3139.1

Tanaka, K. and Raddatz, T., 2011. Correlation between climate sensitivity and aerosol forcing and its implication for the "climate trap". *Climatic Change*, 109, 815-825. https://link.springer.com/article/10.1007/s10584-011-0323-2

---

## Referee Comment (RC1) · Anonymous Referee #1 · 19 Feb 2020

**1 Summary**

Annan et al. (2020) examine the utility of using natural variability of global mean surface temperature (or ocean mixed layer temperature) to constrain equilibrium climate sensitivity. They extend recent work on this topic by using a two layer energy balance model in a "perfect model" framework. They find that the strength of variability-based constraints is substantially weaker when climate sensitivity is large, which is true for both simulations with no external forcing and simulations that use estimates of historical external forcing. For moderate climate sensitivity (2.5 K), the uncertainty in ECS is approximately 4 degrees using information from the entire time series and includ-

ing aerosol forcing uncertainty. For simpler constraints, the uncertainty range is even larger.

This work is a useful expansion of recent literature on this topic. The manuscript is clearly written, though I suggest the authors make a minor modification to the organization, expand their discussion in several places, and consider condensing some figures (or adding a summary figure) to help compare results across the various experiments.

**2 General Comments**

One suggestion to improve the manuscript is to make it easier for the reader to compare across experiments / figures. For example, Figure 1 and 5 are similar and it would be useful to compare all of these results together (perhaps via plotting them on the same axis with color coding in an additional summary figure or grouping or by putting them on a common figure with different panels). It would be similarly useful to intercompare the various posterior estimates (Fig. 2 and 6 as well as 3, 7, and 8). Perhaps plotting lines corresponding to the 5 - 95% CI and a dot for the most likely value value (which would illustrate the skewness) would help compress the figures (though this may run afoul of the Bayesian framework).

There were a few places (noted below) where it would be helpful to more directly compare and discuss this work in the context of other literature. For example, in some places I thought that Cox et al had commented on some issues (e.g., de-trending, two layer models, etc.) and it wasn't immediately clear how to put that work in the context of this manuscript.

I was confused by the $\epsilon$ factor you used as well as the references for the two layer model used here (see below). It would be helpful to clarify some of this in the revised manuscript.
In terms of organization, I thought it would be helpful to include the data (CMIP + HadCRUT) somewhere in the beginning (e.g., a renamed Methods section), rather than introducing with the manuscript's results.

**3 Specific Comments**

Abstract (line 2) and Page 1 / Line 24: In the abstract I wasn't sure which studies you were referring to that tried to constrain ECS with the trend. This might be an oversimplification of these approaches (the Gregory et al. 2002 and Otto et al. 2013 papers cited), since these publications also considered radiative forcing and ocean heat uptake. I believe more recent work by Jimenez-de-la-Cuesta and Mauritsen (2019; doi: 10.1038/s41561-019-0463-y) and Nijsse et al (2020; doi: 10.5194/esd-2019-86) are consistent the abstract language (with caveats that they focus on TCR and a specific time period). I suggest revising this language to reflect the "energy budget constraint" rather than the trend and/or citing these other relevant publications.

Abstract / Line 15: "observed...observational" consider using "inferred from the de-trended observational record"

Page 2 (general comment): Other studies that discuss the utility of variability in understanding the climate response to external forcing include Langen and Alexeev (2005, doi: 10.1029/2005GL024136) and Kirk-Davidoff (2009, doi: 10.5194/acp-9-813-2009). The latter publication seems particularly relevant to the manuscript under review and could be compared to the results here.

Page 2 / Line 21: Cox et al replied (Cox et al, 2018b) to these comments with some analyses relevant to this manuscript. In it, they discuss issues such as the importance of de-trending, their own two layer experiments, and the effect of historic external forcing. Given the relevance to this manuscript (e.g., they two box model results), some of this information could be presented in the introduction or at least compared to the two

layer results shown in this work.

Section 2: Consider describing the CMIP data and HadCRUT observations here

Page 3 / Line 7: In Cox et al (2018b), they did test a two layer model (building off their one-layer model results)

Page 3 / Line 20 and Equation 2: I was confused why $\epsilon$ did not appear in Eq. 2 or why it wasn't absorbed into gamma in both Eqs. 1 and 2. In quickly looking at the Winton et al (2010) paper: don't they use this term in part because it is a one layer model (line 10 and line 20 seem to imply this was a two layer model, but I realize now this may not have been intended)? Can $\epsilon$ be removed here since you explicitly have deep ocean representation? I would appreciate more text justifying / clarifying the purpose of $\epsilon$. It looks like some of what you attribute to Winton et al (2010) should be attributed to Held and Winton (2010, doi: 10.1175/2009JCLI3466.1)?

Page 3 / Line 26: On first read, I thought you had used a value of the average mixed and deep layer depth based on observations. Suggest making this more clear with something like, "We assume a mixed and deep layer depth of 75 and 1000 m, respectively, which are used to calculate the heat capacities (Cm and Cd, respectively) based on ocean coverage of 70% of the planetary surface area."

Page 3 / Line 27 - 29: It would be useful to provide more information about how you chose your parameter values (and later information about how you get the range of plausible values), citing literature relevant to the selection of these values. It was unclear to me why you didn't simply use the mean or median from Geoffrey et al. (2013), for example.

Page 5 / Line 11 - 12: This suggests that you checked this using the two-layer model, consider putting "(not shown)" to indicate that you checked this.

Page 5 / Line 19 - 20: Does detrending (using 55 year windows as in Cox et al) alter the analysis? It seems like it would be reasonable to linearly detrend (as done in the

Cox et al calculation). In the Cox et al reply, they note that it is still important to de-trend unforced simulations using a two layer model.

Page 5 / Line 23 - 29: This is interesting and useful, though I am not sure how to square this with the results presented in Cox et al. (2018). Could you comment more on this? Is this heteroscedasticity included in their estimate of ECS via linear regression? For example, if you varied S to correspond to the 16 models used in Cox et al. (2018), ran a 150 simulation, and performed linear regression ($\Psi$ versus ECS) would the fit be significantly different from what would be obtained using the 1000-year simulations? Or, in another way, is this issue included in the Cox et al (2018) ECS estimate because the increased variance in high ECS models has the effect of making their linear fit between $\Psi$ and ECS more uncertain (and thus contributes to the uncertainty in GCM $\Psi$ values and, in turn, ECS)? Or is the strength of the relationship in Cox et al fortuitous (as suggested by the importance of what GCM simulations are included in the regression as seen in the Po-Chedley et al comment)? Or perhaps the take home message should be that the observed value of $\Psi$ is uncertain. The Cox et al reply (Extended Data Figure 2, top left) suggests that you would generally expect to get a reasonably strong relationship between $\Psi$ and ECS (even though heteroscedasticity should influence their results, too).

Page 5 (Figure 1 discussion): Note that Po-Chedley et al (2018) found that you only recover a strong $\Psi$ - ECS relationship using all of the piControl data and the relationship is weaker with shorter time series. Kirk-Davidoff (2009) also concludes that shorter time series cannot accurately diagnoses climate sensitivity. Of relevance, Nijsse et al (2019, doi: 10.24433/CO.6887733.v1) show that a metric of decadal surface temperature variability (from piControl data) scales with ECS.

Page 6 / Line 31 - 32: It would be useful to provide a reference regarding this point (since the subsequent linguistic calibration was not immediately intuitive).

Page 6 / Line 32: Should this be "Relative" (where it says "Likelihood values can be

read. . ."?

Figure 1 (and others): Suggest adding a legend with "Single layer (gamma = 0)" and "Two layer". In general, it would be helpful to the reader to have a legend on all of the figures (with the possible exception of Fig. 4) and there appears to be plenty of white space to do so.

Figure 2: Suggest "dashed" instead of "dotted." The blue appears purple on my monitor.

Page 8 / Line 2: You could cite Roe and Baker (2007; doi: 10.1126/science.1144735) and perhaps others here.

Page 9 / Line 5: I don't have intuition for what the relative uncertainty should be, but 20% struck me as small (particularly given the later remark that the observational interannual variability looks relatively large compared to the model simulations).

Page 9 / Line 8: Consider using "purple" instead of "blue" so the reader doesn't get confused with the cyan line (at least this appears purple on my screen).

Page 11 / Line 26: Does this correspond to any published values of the aerosol forcing uncertainty? Later, you say that a larger aerosol forcing corresponds to larger ECS, but this range suggests that you do not, by default, consider larger aerosol forcing - is that right? If so, why?

Page 11 / Line 33 onwards: Could it also be that the noise term in the two layer models is too small?

Page 12 / Line 2: At first I didn't understand what 0.13 represented. Consider re-writing as something like: For each of the three simulations, the RMS differences between model output with no internal variability and observations is 0.13 oC.

Page 13 Line 1 (and elsewhere): This record contains more than the 20th century. Consider using a more generic term (like historical) and noting the time period considered. Also consider using "first" instead of "firstly."

Page 13 / Line 4: In Figure 5, you also only show two layer solutions, which is different from Fig. 1 and could be noted here.

Page 13 / Line 18: Should some of this CMIP information be included in the Methods Section (perhaps revised to "Data and Methods")? What RCP scenario was used to extend the historical time series. This would be a good place to cite Taylor (2011, doi: 10.1175/BAMS-D-11-00094.1). There is also some language that is suggested for acknowledging ESGF and modeling groups (https://pcmdi.llnl.gov/mips/cmip5/citation.html). It would also be helpful to say what variable you are using (tas?).

Page 13 / Line 25 - 27 / Figure 5: It is very useful seeing many CMIP5 realizations on this plot. This is a nice illustration of one of your key points (the range of $\Psi$ values can be quite large for a given model).

**4  Grammatical / Other Comments**

Page 1 / Line 23: Should this be *mid* 19th century to the early *21st* century

Page 2 / Line 1: focussed -> focused

Page 2 / Line 2: Suggest: "and this topic" -> "which"

Page 2 / Line 31: Remove "to" in "from to"

Page 3 / Line 7: Suggest changing "equilibrium sensitivity" to "equilibrium climate sensitivity"

Page 6 / Line 16: Insert: "so *we* perform. . ."

Page 9 / Line 25: I was not familiar with "i.i.d." - suggest writing this out.

---

## Referee Comment (RC2) · Anonymous Referee #2 · 20 Feb 2020

General comments:

Annan et al study whether the variability can constrain sensitivity in an idealized two-box model setting. They find it works well for lower values of sensitivity although loses power for higher values (around 5K). They also report that using the forced response in addition can constrain estimates further. It is well written and seems to be technically correct. I have a few questions listed below and some minor suggestions for ease of comparison with previous work.

Specific comments:

[Figure]

- Ranges are reported in the 5-95% interval. It would be great to see the 33% to 66% for comparison with the IPCC ranges particularly in the abstract, as was done in Cox et al (2018).

- Section 3.1, p.6, lines 25-30. Is the increasing uncertainty in estimated sensitivity due to the larger timescales in the higher sensitivity models relative to the length of the time series which is fixed at 150 years? If climate sensitivity scales proportionally with timescale then the 1K sensitivity has 10 times greater effective sampling for a finite length record. Is this what's going on here?

- Section 4, p. 11. There are other deterministic forcing factors uniform in amplitude and phase across all the CMIP GCMs (and the real Earth system) not present in the two-box model simulations. These deterministic forcing factors could further separate and discriminate the model sensitivities in addition to the IPCC annual forcing timeseries. Examples of such factors are the diurnal and seasonal cycles of solar insolation. Even though these forcings (and responses) are averaged over when using annual GMSAT, there might still be traces of it in the responses (at least in the more complex and nonlinear CMIP model responses and the real world) and this could act to further help reveal the sensitivity. This is not considered in the ideal model scenario and would be interesting to test in the two-box.

- section 4, p. 12. Worth noting that there are differences between forcing from well mixed GHGs and aerosol forcings. Well mixed GHGs tend to act uniformly over the globe while aerosol forcing is quite geographical. You're not going to be able to compare the effects of aerosol forcing with well mixed GHG forcing in a box model, at least in a simple way.

Technical corrections:

- line 25: The usual approach to integrating stochastic differential equations is the Euler-Maruyama method to simulate the correct variance on the random variable. If the timestep is 1 unit (as it appears here) it shouldn't make any difference but is worth noting for the reader.

- Table 1: Would be good to get all the parameters in the same units as Geoffroy et al (2013a) for direct comparison (particularly $C_m$ and $C_d$ in W yr /m$^2$/K). I'm aware these are not standard SI units, but just like kWhrs, they are *much* more convenient to calculate with. It would also be good to list the resulting timescale ranges $\tau_f$ and $\tau_s$ for the same reason.

---

## Author Comment (AC1) · 11 May 2020

Thank you for the helpful comments. We will note the stronger relationship to transient response, albeit it is not a primitive parameter in our model.

The likelihood comment was a bit of an aside to our main analysis and we believe posterior pdfs are more relevant to comparisons with other work.

As for the Held/Winton citation, thank you for the clarification which has helped to explain something that has long confused the first author, since Held et al appeared to be specifically citing Winton et al as the source of their model (eqns 9 and 10 in Held

et al).

The other minor errors will be corrected.

---

## Author Comment (AC2) · 11 May 2020

Thank you for your interesting and helpful comments. We believe that although the relationship with Psi is as you note dimensionally inconsistent, this doesn't actually indicate any error as the missing dimensions are effectively included in the constant of proportionality through the implicit assumption of Cox et al that all other model parameters (some of which have dimensions attached) are held fixed. Relating to your comments about detrending, as the stated purpose was to remove the forced trend, we did not perform this step for the unforced simulations. However doing so does not significantly affect our results, a point that will be noted in the revision.

The reason for the inclusion of epsilon in the model is simply that it is widely regarded as a significantly better model for the inclusion of this term. It has very little influence on our results (at least if we replaced gamma*epsilon by gamma in our simulations) as we do not examine deep ocean temperature. The simulations in which S alone varies do provide an end case to the situation of excessive uncertainty in other parameters.

---

## Author Comment (AC3) · 11 May 2020

Thank you for the useful comments. Here we present some initial responses which describe proposed changes to the manuscript.

General comments:

Figures

We thank the reviewer for the suggestion to combine figures, and have combined figs 2 and 6 in a manner that we hope indicates the shapes and compares the spreads (5-95% range, and location of maximum). Figs 3, 7 and 8 are also combined in a similar

manner. We attach one figure to illustrate the proposed approach. We agree with the reviewer that a line and dot is adequate for summarising the information, especially with some distributions still being shown in full. Figure captions and legends will be improved.

However, for figures 1 and 5 it rapidly became messy to plot all of these data on one figure, especially with us now including two different CMIP ensembles on Figure 5 (sufficient data from CMIP6 are available and it seems a useful validation through being published subsequently to our analysis). We will make sure to emphasise their similarities and differences in the text, eg that they are broadly similar albeit with the 20th century simulations generating slightly higher values for psi than the unforced two-layer simulation in Figure 1.

We have also included a brief mention of the HadCRUT data and two CMIP model ensembles in the methods section.

Specific comments

Abstract: citations in the abstract are deprecated by the journal but we will change the wording a little to the deliberately less specific "trends in observational time series" and expanded the introduction "the focus has been on the long-term energy balance as constrained by the warming trend in surface and ocean temperatures"

l15 : done as suggested

Kirk-Davidoff results will also be discussed in relation to ours.

l21: Citation will be added but additional discussion is retained for later section. Given their original theory was based solely on the single layer unforced model we think it is appropriate to focus initially on this situation.

Section 2: Brief description of HadCRUT and two CMIP ensembles added

p3 l 20: epsilon does not play a significant role here; we include it primarily because

this is the standard version of the model that is widely used to mimic GCM behaviour. We now make this point in the manuscript. It is not quite correct to say that it epsilon can be subsumed into the gamma parameter, as its effect on energy balance is more akin to an additional feedback into space, the strength of which depends on the degree of disequilibrium. As for the references here, while the Held et al reference is actually clearer as to the formulation of the model (see their equations 9 and 10 ) they did specifically cite Winton et al as the origin of this approach. We will try to clarify this in the text.

p3 l26 Agreed

l27-9 Will expand. Aim is not to specifically emulate a particular GCM ensemble but just to cover a reasonable range wherein we believe reality could plausibly lie.

p5 l11 agreed (yes we did look at this briefly).

l23-29. Detrending the unforced runs made very little difference. We thought it was more in the spirit of the original derivation to not do this in the main analysis, as the stated purpose was to remove the forced trend which we know to be zero in this instance. Their ordinary least squares analysis makes no assumption of heteroscedasticity, and we believe they must just have been lucky in their set of models (combined to some extent with optimising the parameters of their analysis).

p6 l31-2 this will be reworded.

Figure legends will be added/improved.

p11 26 This scaling considers larger aerosol forcing (alpha >1) just as likely as smaller (alpha <1) and the 95% range is 0-2x the standard value.

p12 l2 noted and text will be improved.

p13 l1 will change to historical

p13 l18 "Data and methods" section will be included as suggested.

[Figure]

[Figure]

**Fig. 1.**

---

## Author Comment (AC4) · 11 May 2020

Thank you for your helpful comments. We will add some extra ranges to the values to help with comparisons.

Section 3.1 Yes in part this is the case. We cannot accurately infer the intrinsic time scale when the time series is not long compared to it.

Sec 4. p11 Possibly, but we don't think such a simple model can address this with much precision. Weak results relating to the annual cycle were obtained by: Knutti, R., Meehl, G., Allen, M. R., & Stainforth, D. A. (2006). Constraining climate sensitivity from

the seasonal cycle in surface temperature, Journal of Climate 19(17), 4224–4233.

Sect 4 yes we will note this. Hemispheric models take advantage of this point.

l25 Noted thank you

Table 1 We will include these values.

---

## Author Response (AR1)

Response to Editor and short comments

Editor comments (E) and author responses (A)

E: I would like to add two additional requests. One apparent simplification is that noise is assumed to be Gaussian. Climate model signals (and observed climate data) has considerable spectral structure (ENSO, PDO, AMOC, etc.). Presumably in these models this is simplified into some modelled split between noise and signal. Could a more realistic noise representation impact the results?

A: While the noise assumption is a little simplistic, it mirrors that made by Cox et al, and is also widely used in the literature. We believe that the heteroscedastic and nonlinear behaviours shown in Fig 5 would be robust to the details of the noise as they arise directly from (a) the reduced damping with large S, and (b) the influence of ocean heat uptake, respectively. The results with CMIP5/6 models also show the relationship between sensitivity and variability to be is weak.

E: It would also be helpful if you could comment if the assumption for the magnitude of the noise (sigma) plays a role in the final results since this appears to be a fixed assumption "generally use the value σ = 0.05".

A: We have add a comment in the manuscript (p 3 line 26). The value of psi scales with the magnitude of the noise but the results are not very sensitive to it. It was chosen as a reasonable compromise between compatibility with model results and surface temperature observations.

In addition, some least significant digits have changed due to sampling variability following the re-ordering of some of the simulations for the new figure arrangement.

Short comment from Nic Lewis (SC) and author responses (A)

SC: [comments about transient response]

A: p6 l30 We note the stronger relationship to transient response, albeit it is not a primitive parameter in our model.

SC: Page 3 line 20: the model in equations (1) and (2) is not the Winton et al (2010)
model: it is the Held et al (2010) model. The Winton et al model, although similar to the
Held et al. model, has a basic difference in that its efficacy parameter ε applies to total
ocean heat uptake, and is found to vary significantly over time in AOGCMs, whereas in
the Held et al model ε only applies to deep ocean heat uptake, as in equation (1), and
is found to fit AOGCM behaviour with (as here) a constant ε value. The cited Geoffroy
et al (2013a) paper uses the Held model, not the Winton model.
R:

A: done

SC Page 4 Table 1: there is a sign error in the default value for the radiative feedback
parameter, λ. The way this parameter is used in equation (1) implies it is negative, but
Table 1 defines it as 3.7/S not −3.7/S.

A: done

SC: Page 13, line 30: the title of Kass and Raftery's 1995 paper is just "Bayes Factors".

A: done

Short comment from  Stephen E. Schwartz

A: No direct changes necessary, though we do mention (p6

l8) that the results are insensitive to detrending or not
in the unforced case.

Response to Anonymous Referee #1
Referee comments (R)
Author comments (A)

R:
1 Summary
Annan et al. (2020) examine the utility of using natural
variability of global mean surface temperature (or ocean
mixed layer temperature) to constrain equilibrium climate
sensitivity. They extend recent work on this topic by
using a two layer energy balance model in a "perfect
model" framework. They find that the strength of
variability-based constraints is substantially weaker
when climate sensitivity is large, which is true for both
simulations with no external forcing and simulations that
use estimates of historical external forcing. For
moderate climate sensitivity (2.5 K), the uncertainty in
ECS is approximately 4 degrees using information from the
entire time series and including aerosol forcing
uncertainty. For simpler constraints, the uncertainty
range is even larger. This work is a useful expansion of
recent literature on this topic. The manuscript is
clearly written, though I suggest the authors make a
minor modification to the organization, expand their
discussion in several places, and consider condensing
some figures (or adding a summary figure) to help compare
results across the various experiments.

2 General Comments
One suggestion to improve the manuscript is to make it
easier for the reader to compare across experiments /
figures. For example, Figure 1 and 5 are similar and it
would be useful to compare all of these results together
(perhaps via plotting them on the same axis with color
coding in an additional summary figure or grouping or by
putting them on a common figure with different panels).
It would be similarly useful to intercompare the various

posterior estimates (Fig. 2 and 6 as well as 3, 7, and 8). Perhaps plotting lines corresponding to the 5 - 95% CI and a dot for the most likely value value (which would illustrate the skewness) would help compress the figures (though this may run afoul of the Bayesian framework).

A: Thanks for the suggestions. As per our initial response in the open review, we have combined figs 2 with 6, also 3, 7 and 8 and think the changes are an improvement. Figure captions and legends have been edited appropriately. We draw attention in the text to differences between figs 1 and 5.

R:
There were a few places (noted below) where it would be helpful to more directly compare and discuss this work in the context of other literature. For example, in some places I thought that Cox et al had commented on some issues (e.g., de-trending, two layer models, etc.) and it wasn't immediately clear how to put that work in the context of this manuscript.
I was confused by the  factor you used as well as the references for the two layer model used here (see below). It would be helpful to clarify some of this in the revised manuscript.

A: We didn't want to focus too directly on the details of Cox et al and the various comments/responses, as we consider the analysis using the full time series to provide a more compelling result that obviates a detailed investigation of their approach as it puts a strict bound on the potential for variability, however it it is analysed, to provide a constraint. The "garden of forking paths" is a very clear danger when a large number of choices can be made in the analysis procedure, and therefore such choices must be supported by an underlying theoretical basis. However we have added discussion of Kirk-Davidoff and made a number of other changes described in more detail below.

R:

In terms of organization, I thought it would be helpful to include the data (CMIP + HadCRUT) somewhere in the beginning (e.g., a renamed Methods section), rather than introducing with the manuscript's results.

A: Subsection on Additional Data to describe CMIP5/6 and HadCRUT data has been included.

R:
3 Specific Comments
Abstract (line 2) and Page 1 / Line 24: In the abstract I wasn't sure which studies you were referring to that tried to constrain ECS with the trend. This might be an oversimplification of these approaches (the Gregory et al. 2002 and Otto et al. 2013 papers cited), since these publications also considered radiative forcing and ocean heat uptake. I believe more recent work by Jimenez-de-la-Cuesta and Mauritsen (2019; doi:10.1038/s41561-019-0463-y) and Nijsse et al (2020; doi: 10.5194/esd-2019-86) are consistent the abstract language (with caveats that they focus on TCR and a specific time period). I suggest revising this language to reflect the "energy budget constraint" rather than the trend and/or citing these other relevant publications.

A: Citations in the abstract are deprecated by the journal but we have changed the wording a little to the deliberately less specific "trends in observational time series" and expanded the introduction to read "energy balance as constrained by the warming trend in atmospheric and oceanic temperatures "

R:
Abstract / Line 15: "observed. . .observational" consider using "inferred from the detrended observational record"

A: Done as suggested

R:
Page 2 (general comment): Other studies that discuss the

utility of variability in understanding the climate response to external forcing include Langen and Alexeev (2005, doi:10.1029/2005GL024136) and Kirk-Davidoff (2009, doi: 10.5194/acp-9-813-2009).The latter publication seems particularly relevant to the manuscript under review and could be compared to the results here.

A: Kirk-Davidoff citation added, and mentioned again p6 l17

R:
Page 2 / Line 21: Cox et al replied (Cox et al, 2018b) to these comments with some analyses relevant to this manuscript. In it, they discuss issues such as the importance of de-trending, their own two layer experiments, and the effect of historic external forcing. Given the relevance to this manuscript (e.g., they two box model results), some of this information could be presented in the introduction or at least compared to the two layer results shown in this work.

A: Cox 2018b Citation has been added but additional discussion is retained for Section 4. Given their original (2018a) result was justified solely on the single layer unforced model we think it is appropriate to focus initially on this situation.

R:
Section 2: Consider describing the CMIP data and HadCRUT observations here

A: Brief description of HadCRUT and two CMIP ensembles added.

R:
Page 3 / Line 7: In Cox et al (2018b), they did test a two layer model (building off their
one-layer model results)

A: No change needed (Cox 2018b is cited and discussed elsewhere).

R:
Page 3 / Line 20 and Equation 2: I was confused why epsilon did not appear in Eq. 2 or why it wasn't absorbed into gamma in both Eqs. 1 and 2. In quickly looking at the Winton et al (2010) paper: don't they use this term in part because it is a one layer model (line 10 and line 20 seem to imply this was a two layer model, but I realize now this may not have been intended)? Can epsilon be removed here since you explicitly have deep ocean representation? I would appreciate more text justifying / clarifying the purpose of epsilon. It looks like some of what you attribute to Winton et al (2010) should be attributed to Held and Winton (2010, doi: 10.1175/2009JCLI3466.1)?

A: Epsilon does not play a significant role here; we include it primarily because this is the standard version of the model that is widely used to mimic GCM behaviour. We now make this point in the manuscript. It is not quite correct to say that it epsilon can be subsumed into the gamma parameter, as its effect on energy balance is more akin to an additional feedback into space, the strength of which depends on the degree of disequilibrium. As for the references here, while the Held et al reference is actually clearer as to the formulation of the model (see their equations 9 and 10 ) they did specifically cite Winton et al as the origin of this approach. We now cite both papers (l12).

R:
Page 3 / Line 26: On first read, I thought you had used a value of the average mixed and deep layer depth based on observations. Suggest making this more clear with something like, "We assume a mixed and deep layer depth of 75 and 1000 m, respectively, which are used to calculate the heat capacities (Cm and Cd, respectively) based on ocean coverage of 70% of the planetary surface

area."

A: Done

R:
Page 3 / Line 27 - 29: It would be useful to provide more information about how you chose your parameter values (and later information about how you get the range of plausible values), citing literature relevant to the selection of these values. It was unclear to me why you didn't simply use the mean or median from Geoffrey et al. (2013), for example.

A: Aim is not to specifically emulate a particular GCM ensemble but just to cover a reasonable range wherein we believe reality could plausibly lie. Added sentence to explain, "Our aim here is not specifically to replicate or mimic this ensemble but to allow for a reasonable range of parameter values."

R:
Page 5 / Line 11 - 12: This suggests that you checked this using the two-layer model,
consider putting "(not shown)" to indicate that you checked this.

A: done. Yes we did look at this briefly.

R:
Page 5 / Line 19 - 20: Does detrending (using 55 year windows as in Cox et al) alter the analysis? It seems like it would be reasonable to linearly detrend (as done in the Cox et al calculation). In the Cox et al reply, they note that it is still important to de-trend unforced simulations using a two layer model.

A: We did test and detrending the unforced runs made very little difference. We thought it was more in the spirit of the original derivation to not do this in the main analysis, as the stated purpose was to remove the forced trend which we know to be zero in this instance.

R:
Page 5 / Line 23 - 29: This is interesting and useful, though I am not sure how to square this with the results presented in Cox et al. (2018). Could you comment more on this? Is this heteroscedasticity included in their estimate of ECS via linear regression? For example, if you varied S to correspond to the 16 models used in Cox et al. (2018), ran a 150 simulation, and performed linear regression (Ψ versus ECS) would the fit be significantly different from what would be obtained using the 1000-year simulations? Or, in another way, is this issue included in the Cox et al (2018) ECS estimate because the increased variance in high ECS models has the effect of making their linear fit between Ψ and ECS more uncertain (and thus contributes to the uncertainty in GCM Ψ values and, in turn, ECS)? Or is the strength of the relationship in Cox et al fortuitous (as suggested by the importance of what GCM simulations are included in the regression as seen in the Po-Chedley et al comment)? Or perhaps the take home message should be that the observed value of Ψ is uncertain. The Cox et al reply (Extended Data Figure 2, top left) suggests that you would generally expect to get a reasonably strong relationship between Ψ and ECS (even though heteroscedasticity should influence their results, too).

A: Their ordinary least squares analysis makes no assumption of (and therefore does not account for) heteroscedasticity, and we believe they must just have been lucky in their set of models (combined to some extent with optimising various choices during their analysis). Our analysis using the exact likelihood renders the entire debate moot. Furthermore, the failure of their approach for CMIP6, and the significantly different relationship exhibited by these models, is additional evidence that their original result was unreliable.

We now also mention the difference between the CMIP ensembles in the conclusions.

R:
Page 5 (Figure 1 discussion): Note that Po-Chedley et al (2018) found that you only recover a strong $\Psi$ - ECS relationship using all of the piControl data and the relationship is weaker with shorter time series. Kirk-Davidoff (2009) also concludes that shorter time series cannot accurately diagnoses climate sensitivity. Of relevance, Nijjse et al (2019, doi: 10.24433/CO.6887733.v1) show that a metric of decadal surface temperature variability (from piControl data) scales with ECS.

A: Po-Chedley and Kirk-Davidoff cited.

R:
Page 6 / Line 31 - 32: It would be useful to provide a reference regarding this point (since the subsequent linguistic calibration was not immediately intuitive).

A: This has been be reworded. It is intended as a literal translation of what P(psi|S) means, however.

R:
Page 6 / Line 32: Should this be "Relative" (where it says "Likelihood values can be
read. . ."?

A: yes, text changed

R:
Figure 1 (and others): Suggest adding a legend with
"Single layer (gamma = 0)" and
"Two layer". In general, it would be helpful to the reader to have a legend on all of the
figures (with the possible exception of Fig. 4) and there appears to be plenty of white
space to do so.

A: Legend added here (and also to other figures).

R:
Figure 2: Suggest "dashed" instead of "dotted." The blue appears purple on my monitor.
Page 8 / Line 2: You could cite Roe and Baker (2007; doi: 10.1126/science.1144735)
and perhaps others here.

A: Changed to dashed lines

R:
Page 9 / Line 5: I don't have intuition for what the relative uncertainty should be, but
20% struck me as small (particularly given the later remark that the observational interannual variability looks relatively large compared to the model simulations).

A: Observational variability has a large component of observational uncertainty, so this must provide an upper bound on internal variability.

R:
Page 9 / Line 8: Consider using "purple" instead of "blue" so the reader doesn't get
confused with the cyan line (at least this appears purple on my screen).

A: We think this colour scheme makes sense, using blue, cyan, magenta and green, with dashed lines representing the results for multiple uncertain parameters. We have added a legend which should help.

R:
Page 11 / Line 26: Does this correspond to any published values of the aerosol forcing
uncertainty? Later, you say that a larger aerosol forcing corresponds to larger ECS,
but this range suggests that you do not, by default, consider larger aerosol forcing - is
that right? If so, why?

A: This scaling considers larger aerosol forcing (alpha >1) just as likely as smaller (alpha <1) and the 95% range is 0-2x the standard value, which we consider to include all reasonable estimates.

R:
Page 11 / Line 33 onwards: Could it also be that the noise term in the two layer models
is too small?

A: This is possible and we've changed the text to mention this.

R:
Page 12 / Line 2: At first I didn't understand what 0.13 represented. Consider re-writing as something like: For each of the three simulations, the RMS differences between model output with no internal variability and observations is 0.13 oC.

A: Done

R:
Page 13 Line 1 (and elsewhere): This record contains more than the 20th century. Consider using a more generic term (like historical) and noting the time period considered. Also consider using "first" instead of "firstly."

A: Changed to historical. Also changed firstly to first.

R:
Page 13 / Line 4: In Figure 5, you also only show two layer solutions, which is different
from Fig. 1 and could be noted here.

A: noted in text at the start of Section 4.1

R:
Page 13 / Line 18: Should some of this CMIP information be included in the Methods Section (perhaps revised to

"Data and Methods")? What RCP scenario was used to extend the historical time series. This would be a good place to cite Taylor (2011, doi: 10.1175/BAMS-D-11-00094.1). There is also some language that is suggested for acknowledging ESGF and modeling groups (https://pcmdi.llnl.gov/mips/cmip5/citation.html). It would also be helpful to say what variable you are using (tas?).

A: "Additional data" section 2.3 has been included as suggested.

R:
Page 13 / Line 25 - 27 / Figure 5: It is very useful seeing many CMIP5 realizations on this plot. This is a nice illustration of one of your key points (the range of Ψ values can be quite large for a given model).

A: Thank you

R:
4 Grammatical / Other Comments
Page 1 / Line 23: Should this be *mid* 19th century to the early *21st* century
Page 2 / Line 1: focussed -> focused
Page 2 / Line 2: Suggest: "and this topic" -> "which"
Page 2 / Line 31: Remove "to" in "from to"
Page 3 / Line 7: Suggest changing "equilibrium sensitivity" to "equilibrium climate sensitivity"
Page 6 / Line 16: Insert: "so *we* perform. . ."
Page 9 / Line 25: I was not familiar with "i.i.d." - suggest writing this out.

A: All done as suggested.

Response to Anonymous Referee #2

A:
Thank you for your helpful comments.

R:

General comments:
Annan et al study whether the variability can constrain sensitivity in an idealized twobox model setting. They find it works well for lower values of sensitivity although loses power for higher values (around 5K). They also report that using the forced response in addition can constrain estimates further. It is well written and seems to be technically correct. I have a few questions listed below and some minor suggestions for ease of comparison with previous work.

R:
Specific comments:
Ranges are reported in the 5-95% interval. It would be great to see the 33%
to 66% for comparison with the IPCC ranges particularly in the abstract, as was
done in Cox et al (2018).

A: We have added some extra ranges to the values for the results with the full time series (our most optimistic scenario) to help with comparisons. (IPCC likely range is 17-83%, which we assume is what the reviewer meant). End of Section 4.2

R:
Section 3.1, p.6, lines 25-30. Is the increasing uncertainty in estimated sensitivity due to the larger timescales in the higher sensitivity models relative to the length of the time series which is fixed at 150 years? If climate sensitivity scales proportionally with timescale then the 1K sensitivity has 10 times greater effective sampling for a finite length record. Is this what's going on here?

A: Section 3.1 Yes in part this is the case. We cannot accurately infer the intrinsic time scale when the time series is not long compared to it.

R:

Section 4, p. 11. There are other deterministic forcing factors uniform in amplitude and phase across all the CMIP GCMs (and the real Earth system) not present in the two-box model simulations. These deterministic forcing factors could further separate and discriminate the model sensitivities in addition to the IPCC annual forcing timeseries. Examples of such factors are the diurnal and seasonal cycles of solar insolation. Even though these forcings (and responses) are averaged over when using annual GMSAT, there might still be traces of it in the responses (at least in the more complex and nonlinear CMIP model responses and the real world) and this could act to further help reveal the sensitivity. This is not considered in the ideal model scenario and would be interesting to test in the two-box.

A: Possibly, but we don't think such a simple model can address this with much precision. Weak results relating to the annual cycle were obtained by: Knutti, R., Meehl, G., Allen, M. R., & Stainforth, D. A. (2006). Constraining climate sensitivity from the seasonal cycle in surface temperature, Journal of Climate 19(17), 4224–4233.

R:
section 4, p. 12. Worth noting that there are differences between forcing from well mixed GHGs and aerosol forcings. Well mixed GHGs tend to act uniformly over the globe while aerosol forcing is quite geographical. You're not going to be able to compare the effects of aerosol forcing with well mixed GHG forcing in a box model, at least in a simple way.

A: Sect 4 yes we now note this at the top of p14 (and also cite Aldrin et al as an example). Hemispheric models take advantage of this point.

R:
Technical corrections:
line 25: The usual approach to integrating stochastic

differential equations is the Euler-Maruyama method to simulate the correct variance on the random variable. If the timestep is 1 unit (as it appears here) it shouldn't make any difference but is worth noting for the reader.

A: We selected the noise sampling deliberately to achieve the desired variance in temperatures.

R:
Table 1: Would be good to get all the parameters in the same units as Geoffroy et al (2013a) for direct comparison (particularly $C_m$ and $C_d$ in W yr /m2 /K). I'm aware these are not standard SI units, but just like kWhrs, they are much more convenient to calculate with. It would also be good to list the resulting timescale ranges $\tau_f$ and $\tau_s$ for the same reason.

A: Table 1 We have added the capacities in these alternative units.

[revised manuscript text omitted]